# Neural Foundations of Mental Simulation: Future Prediction of Latent Representations on Dynamic Scenes

**Aran Nayebi**[1,*]**, Rishi Rajalingham**[1,4]**, Mehrdad Jazayeri**[1,2]**, and Guangyu Robert Yang**[1,2,3]

[1]McGovern Institute for Brain Research, MIT; Cambridge, MA 02139
[2]Department of Brain and Cognitive Sciences, MIT; Cambridge, MA 02139
[3]Department of Electrical Engineering and Computer Science, MIT; Cambridge, MA 02139
[4]Reality Labs, Meta; 390 9th Ave, New York, NY 10001
[*]Correspondence: `anayebi@mit.edu`

## Abstract

Humans and animals have a rich and flexible understanding of the physical world, which enables them to infer the underlying dynamical trajectories of objects and events, plausible future states, and use that to plan and anticipate the consequences of actions. However, the neural mechanisms underlying these computations are unclear. We combine a goal-driven modeling approach with dense neurophysiological data and high-throughput human behavioral readouts that contain thousands of comparisons to directly impinge on this question. Specifically, we construct and evaluate several classes of sensory-cognitive networks to predict the future state of rich, ethologically-relevant environments, ranging from self-supervised end-to-end models with pixel-wise or object-slot objectives, to models that future predict in the latent space of purely static image-pretrained or dynamic video-pretrained foundation models. We find that "scale is *not* all you need", and that many state-of-the-art machine learning models fail to perform well on our neural and behavioral benchmarks for future prediction. In fact, only one class of models matches these data well overall. We find that neural responses are currently best predicted by models trained to predict the future state of their environment in the *latent* space of pretrained foundation models optimized for *dynamic* scenes in a self-supervised manner. These models also approach the neurons' ability to predict the environmental state variables that are visually hidden from view, despite not being explicitly trained to do so. Finally, we find that not all foundation model latents are equal. Notably, models that future predict in the latent space of video foundation models that are optimized to support a *diverse* range of egocentric sensorimotor tasks, reasonably match *both* human behavioral error patterns and neural dynamics across all environmental scenarios that we were able to test. Overall, these findings suggest that the neural mechanisms and behaviors of primate mental simulation have strong inductive biases associated with them, and are thus far most consistent with being optimized to future predict on *reusable* visual representations that are useful for Embodied AI more generally.

## 1   Introduction

Within the span of a couple seconds, we are able to draw rich inferences and make predictions about novel scenes [Smith and Vul, 2013, Battaglia et al., 2013]. A dominant cognitive theory has been that the brain builds mental models of the physical world, using those models to make inferences about the

future state of its environment [Craik, 1943]. In the past decade, this hypothesis has been supported by comparisons of human behavior to computational models which predict what will happen next in physical scenarios via forward simulations resembling those of game engines in modern video games [Battaglia et al., 2013, Hamrick et al., 2016, Ullman et al., 2017]. Both neuroimaging work in humans [Zacks, 2008, Fischer et al., 2016, Schwettmann et al., 2019, Pramod et al., 2022] and recent electrophysiological work in monkeys [Rajalingham et al., 2022a,b] has further provided evidence for the neurobiological basis of mental simulations in the frontoparietal network (FPN) of primates, a large-scale network consisting of several interacting brain regions. In this work, we make progress towards understanding the neural and behavioral mechanisms of mental simulation by constructing models that perform this behavior in rich, naturalistic environments. Specifically, we aim to determine what inductive biases (in the form of a loss function, architecture class, and pretraining environment) enable the brain to generally perform mental simulation *across* a range of environments and scenarios, from unstructured, continuous sensory inputs. In particular, we assess not only model generalization to novel, high-variation examples within the same environment, but also *structural generalization* to new environments and scenarios altogether.

Predicting the physical dynamics of environments is also critical to progress in Embodied AI. One common paradigm for learning these dynamics has been as a next frame prediction problem via a pixel-wise loss [Villegas et al., 2019, Wu et al., 2021, Babaeizadeh et al., 2021, Nash et al., 2022]. These losses emphasize prioritizing accurate prediction of every detail of a given scene's dynamics. However, fine-grained prediction of upcoming video frames would require near-perfect knowledge of the world's physical state (akin to Laplace's Demon), which may explain the observation why many of these models tend to *underfit* in high variation, naturalistic visual environments, with recent efforts aimed at scaling these methods up primarily by increasing their parameter count [Dasari et al., 2019, Babaeizadeh et al., 2021]. It is therefore unclear how much these resultant learned representations are able to successfully capture general physical understanding.

Another recent class of approaches involves the design of visual "foundation models" [Bommasani et al., 2021], trained on large amounts of webscale images and egocentric videos to develop an implicit representation of the world, that can then be deployed to downstream robotic manipulation tasks [Nair et al., 2022, Ma et al., 2023, Majumdar et al., 2023]. Of course, these models are not directly designed to do explicit physical simulation, but we equip them with a forward dynamics model that can be rolled out for an arbitrary number of timesteps. We ask whether such dynamically-equipped foundation models have learned physical knowledge by evaluating their generalization both to new scenarios and environments, *and* whether their representations bear any similarity to humans and non-human primates performing the same tasks?

In particular, we find strong constraints on primate mental simulation, especially when examining generalization within and across diverse environments. Our core result is that a small class of models best match primate frontal cortex neural dynamics while the animal plays a ball interception task in which the ball trajectory is partially occluded prior to hitting the paddle ("Mental-Pong"), developed previously by Rajalingham et al. [2022a]. Overall, one model class can match both neural response dynamics *and* human behavioral patterns reasonably well – namely, dynamics that are optimized to future predict in the latent space of VC-1, a video foundation model pretrained on the largest variety of egocentric sensorimotor settings overall. We therefore currently observe a tight correspondence between the ability to predict fine-grained neural and behavioral responses for the mental simulation phenomenon, and developing useful representations for Embodied AI more generally.

## 2   Related Work

Mental simulations have been studied at the level of neural activity only very recently. Prior human neuroimaging studies [Zacks, 2008, Fischer et al., 2016, Pramod et al., 2022] in the past decades showed elevated levels of blood-oxygen-level-dependent (BOLD) signal to mental simulation, although they do not have the required resolution to verify that these dynamics are actually represented in underlying neural activity. Rajalingham et al. [2022b] was the first study to show that neural dynamics recorded from macaque dorsomedial frontal cortex (DMFC), track the occluded ball by comparing these dynamics to Recurrent Neural Networks (RNNs) that simulate the occluded ball's position in Mental-Pong, and finding that they better match these dynamics than RNNs that only perform ball endpoint prediction. However, monkeys can perform these tasks without substantial training, suggesting that they are already equipped with the necessary neural foundations for mental

simulation in this environment. Therefore, we aim to also build networks that are not explicitly trained on Mental-Pong itself, but are tasked to *generalize* to this novel setting as a test of their general understanding of physical scene dynamics – chiefly developed through three factors: their architecture, optimization objective, and pretraining on a naturalistic environment.

Additionally, we constrain our models by evaluating them against high-throughput human behavioral data (from Bear et al. [2021]) in more naturalistic, 3D environments than Mental-Pong alone, which goes beyond prior behavioral studies that either rely on a narrow range of physical scenarios [Shepard and Metzler, 1971, Cooperau and Shepard, 1973], such as block towers with several cubes of different colors [Groth et al., 2018, Li et al., 2016], or 2D environments that may not generalize to the real world [Bakhtin et al., 2019]. A key challenge to addressing these questions is a common standard to evaluating the everyday physical scene understanding and neural predictivity of these models, especially since they are usually trained on vastly different scenarios and input types. Towards this end, we require models to operate under similar constraints as the brain, namely **(i)** to take in unstructured visual inputs across a range of physical phenomena, **(ii)** to generate physical predictions for each scene (i.e. producing "behavioral outputs"), and **(iii)** to consist of internal units that can be compared to biological units (i.e. containing "artificial neurons").

Taken together, these three requirements encompass a large class of functionally reasonable hypotheses that we call "sensory-cognitive networks", and includes the two broad approaches mentioned in §1. However, they do exclude some approaches – for example, particle-based graph neural network dynamics predictors [Battaglia et al., 2016, 2018, Li et al., 2019, Sanchez-Gonzalez et al., 2020] that take the ground truth simulator state as input (which may not be readily available in real-world situations, failing to satisfy requirement (i)) or probabilistic programs [Battaglia et al., 2013, Hamrick et al., 2016, Ullman et al., 2017] (which fail to satisfy requirements (i) and (iii)).

Nonetheless, we believe these latter approaches from cognitive science are a useful guide for building improved models from pixels that satisfy the three requirements – especially in terms of assessing whether the prediction problem lies at the level of vision, or the dynamics that interfaces with it. For example, [Bear et al., 2021, Figure 5] demonstrated that particle-based graph neural networks (e.g. DPI-Net [Li et al., 2019]) with access to the ground truth simulator state approach human-level physical predictions on the OCP task that we consider in Section 4 across a diverse range of scenarios. This observation indicates that the onus mainly rests on learning a good visual representation, as these models assume perfect observability of physical dynamics. This further motivates our enforcement of requirement (i) as being a worthwhile challenge, as well as our latent future prediction approach of "factorizing" the mental simulation problem into separately pretrained vision and dynamics modules.

## 3 Methods

To tackle this question, we took a hypothesis-driven approach and built sensory-cognitive networks that performed mental simulations of their environment in a variety of different ways (schematized in Figure 1). Specifically, we tasked models to operate on the Physion dataset [Bear et al., 2021], a large-scale video dataset that focuses on everyday physical understanding, consisting of eight different scenarios in a simulated Unity3D-based environment (the ThreeDWorld simulator [Gan et al., 2021]) with roughly 2,000 scenes each. Models are pretrained on all eight scenarios of Dominoes, Support, Collide, Contain, Drop, Link, Roll, and Drape; which together cover many scenes involving rigid and soft bodies.

We additionally pretrain a subset of models on Kinetics-700 [Carreira et al., 2019], which consists of over 500,000 training videos from real-world (rather than simulated) scenes from 700 different action categories, in order to assess whether a different dataset of increased scale is beneficial or not, relative to Physion.

We consider models from several sensory-cognitive hypothesis classes:

1. **End-to-end self-supervised future prediction:**
   (a) **Pixel-wise:** This class of models consists of three parts: encoder, dynamics, and decoder; which are altogether pretrained end-to-end with a pixel-wise loss to predict the next frame. We consider a current state-of-the-art model in this model family, FitVid [Babaeizadeh et al., 2021], and we pretrain it on Physion (with and without temporal augmentations such as RandAugment [Cubuk et al., 2020]); along with an

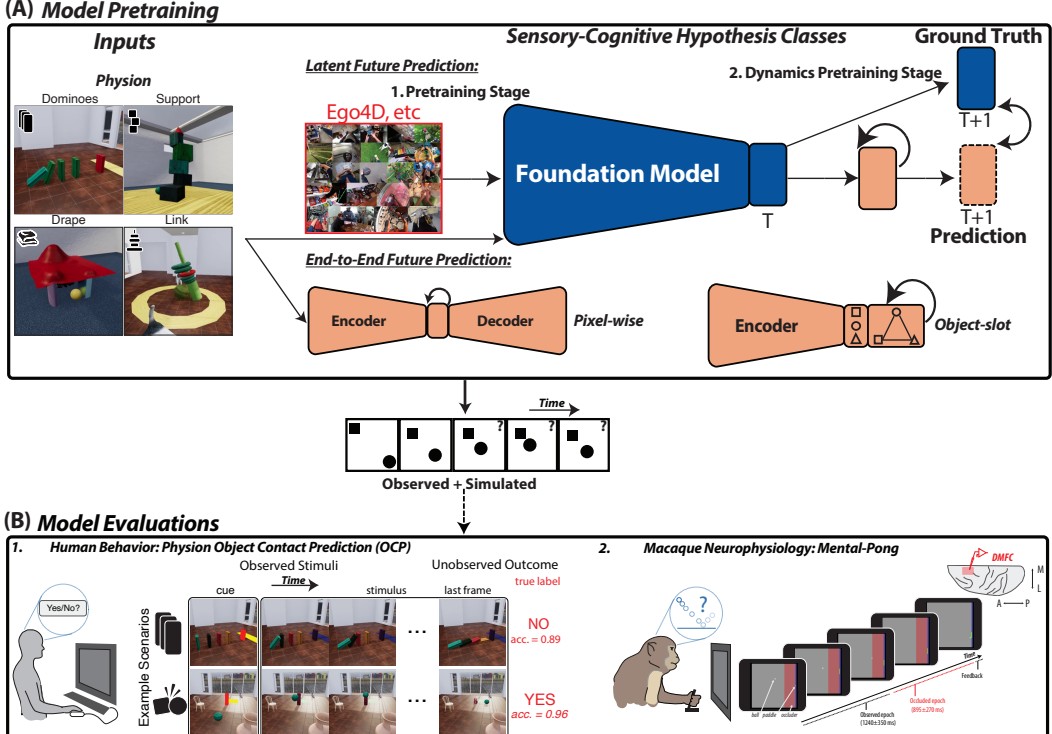

Figure 1: **Model Pretraining and Evaluation Pipeline:** We built and evaluate sensory-cognitive networks for mental simulation. **(A)** Models are pretrained to predict the future state of naturalistic environments in a variety of ways. Their dynamics are then compared to high-throughput human behavioral judgements and dense primate neurophysiological response dynamics. Model pretraining can be (1) two-stage, or (2) end-to-end. For the two-stage latent future prediction models, there is first visual module pretraining (blue), then dynamics module pretraining on Physion (peach). Across all models, the dynamics modules are pretrained on the *same* dataset (Physion), and then evaluated against neural and behavioral data with model weights fixed. **(B)** Evaluations are multi-fold: (1) Comparison to human behavior in OCP on held-out scenes across pretrained Physion scenarios. (2) Comparison to neural activity in Mental-Pong, an out-of-distribution environment that *none* of the models were pretrained on.

earlier variant, SVG [Denton and Fergus, 2018], that can be additionally trained at larger image scales ($128 \times 128$ rather than $64 \times 64$ pixels).

(b) **Object-slot:** This class of models is based on the Contrastive Learning of Structured World Models framework ("C-SWM" [Kipf et al., 2020]) which contrastively learns more structured object-slot latent states. It consists of an encoder (whose size we vary: "small", "medium", and "large") that outputs a fixed number of object slots $N = 10$, and a graph neural network forward dynamics module that operates on this representation. Since the dynamics module itself does not have temporal dependencies (only spatial), we enable temporal dependencies by passing frames in the input channel dimension of the encoder, as a sliding window of temporal dimension of $T$ context frames to predict the object-slot latent at timestep $T + 1$. This model family is *one* instantiation of the cognitive theory that humans tend to reason about scenes with regards to objects and their relations [Baillargeon et al., 1985, Spelke, 1990, Spelke and Kinzler, 2007]. These Gestalt principles therefore may provide physical knowledge that are better-positioned to generalize than more fine-grained (pixel-wise) prediction – a hypothesis that we explicitly evaluate on high-throughput neural and behavioral data, in what follows.

2. **Latent self-supervised future prediction:** These models consist of two parts: an encoder $\mathcal{E}$ that produces a latent state space $h$ and a dynamics module $\mathcal{D}$ that predicts the future state

purely in the latent space $h$ of $\mathcal{E}$. $\mathcal{E}$ is fixed and pretrained to perform a challenging vision task ("foundation model"), through which it learns a partial, implicit representation of the physical world. However, since $\mathcal{E}$ is *not* pretrained to do explicit physical simulation, we train additional dynamics $\mathcal{D}$ on Physion, that can later be "rolled out" an arbitrary number of steps. More concretely, given $T$ context frames, $\mathcal{E}$ produces the latent sequence $h_{1:T}$, from which $\mathcal{D}$ is trained to predict $h_{T+1}$. $\mathcal{D}$ is relatively simple, being either an LSTM [Hochreiter and Schmidhuber, 1997] or a continuous-time RNN (CTRNN) [Miller and Fumarola, 2012]; or "No Dynamics" as a control, which always outputs the last context frame latent $h_T$ of $\mathcal{E}$.

We consider a variety of foundation encoders, divided into two primary classes based on their type of large-scale pretraining dataset:

(a) **Image Foundation Models (Static scenes):**
- Standard Convolutional Neural Networks (CNNs) VGG16 [Simonyan and Zisserman, 2014] and ResNet-50 [He et al., 2016] pretrained on ImageNet with a supervised categorization objective.
- Vision Transformer (ViT) [Dosovitskiy et al., 2021] based architectures such as DeiT [Touvron et al., 2021] and DINO [Caron et al., 2021] pretrained on ImageNet with a self-supervised objective.
- DINOv2 [Oquab et al., 2023], which is a very recent ViT pretrained on a larger curated dataset called LVD-142M (142 million images).
- CLIP [Radford et al., 2021], which is a ViT pretrained on 400 million image-text pairs curated from the Internet (250 times larger than ImageNet).

(b) **Video Foundation Models (Dynamic scenes):**
- R3M [Nair et al., 2022], which is a ResNet-50 architecture pretrained with a temporally-contrastive video-language alignment objective on 5 million frames from a subset of the recent large-scale Ego4D human video dataset [Grauman et al., 2022].
- VIP [Ma et al., 2023], which is a ResNet-50 architecture pretrained with a goal-conditioned value function objective on 5 million frames from a subset of Ego4D.
- VC-1 [Majumdar et al., 2023], which is a very recent ViT pretrained on 7 different egocentric video sources (over 5.6 million frames, including Ego4D) relevant to sensorimotor skills, using a self-supervised masked autoencoding (MAE) [He et al., 2022] objective. While MAE is pretrained on individual frames, the statistics of the frames are informed by egocentric motion, unlike webscale images.

In total, these networks encompass both qualitatively distinct hypotheses (pixel-wise vs. object-slot vs. latent future prediction), alongside several variations *within* each hypothesis class. This combination of diverse networks allows us to potentially differentiate hypothesis classes across a range of functionally capable instantiations of each hypothesis.

## 4 Comparison to Human Physical Judgements

### 4.1 OCP task evaluation

In the object contact prediction (OCP) task [Bear et al., 2021], each evaluation scenario involves a red "agent" object and a yellow "patient" object (which did *not* appear during model pretraining). Both humans and models are tasked to predict the probability that they will come into contact. This prediction requires understanding of the relevant physical phenomenon in the given scenario, corresponding to a higher-order readout of the underlying scene dynamics.

### 4.2 End-to-end pixel-wise future predictors best predict human behavior in the *same* environment

The model comparison results are summarized in Figure 2. In Figure 2A, we examine held-out scene accuracy of models across all eight scenarios in the OCP task, where chance performance is 50%. We can see that humans are quite reliable on this task, attaining 74.04% average held-out accuracy across scenarios (grey horizontal line). Furthermore, the best models that approach human accuracy are models that are pretrained end-to-end with pixel-wise losses (FitVid and SVG). Despite

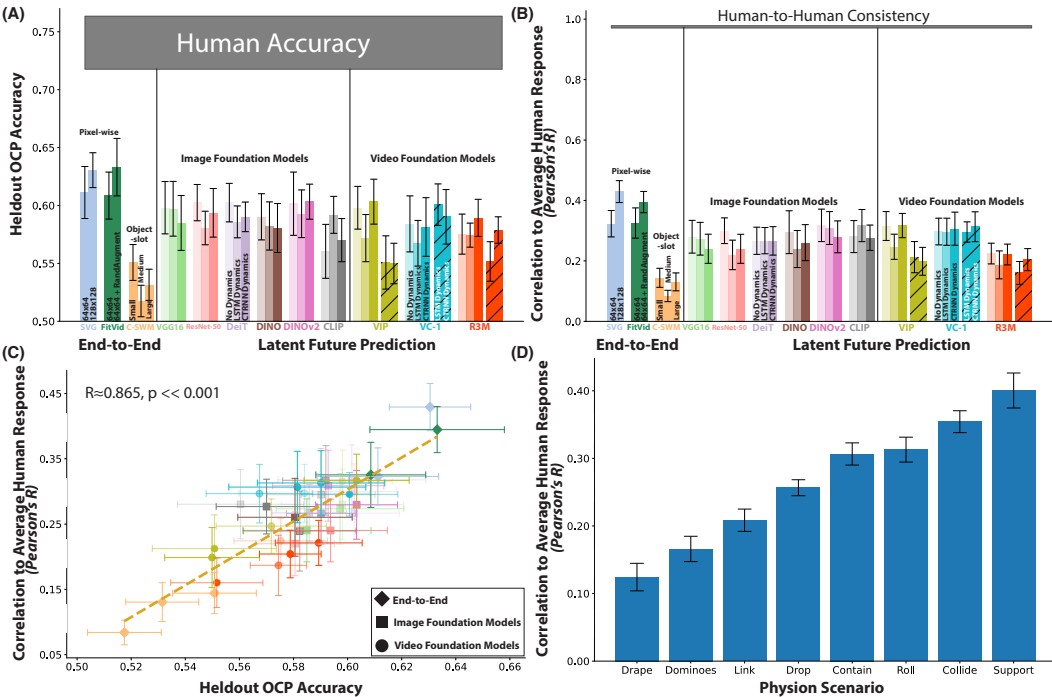

Figure 2: **Model Comparisons to Human Physical Judgements:** **(A)** Each model is compared to binary ("yes/no") accuracy on the OCP task (chance is 50%). Grey horizontal bar represents the human accuracy on this task. Mean and s.e.m. across all eight Physion scenarios, weighted by number of scenes in each scenario. Hatched bars represent models pretrained on the Kinetics-700 dataset, rather than on Physion. **(B)** Each model is compared to human subject judgement probabilities for the OCP task, via Pearson's correlation from the Logistic Regression classifier trained on each of the model dynamics. Grey is the correlation of human judgement probabilities to each other. Mean and s.e.m. across all eight Physion scenarios, weighted by number of scenes in each scenario. **(C)** A model's match to human judgement probabilities is strongly correlated with its OCP task accuracy. **(D)** Per scenario model comparisons to human subject judgement probabilities for the OCP task. Mean and s.e.m. across all 41 models.

their differences in architecture, both FitVid and SVG are comparable to one another in their OCP test set accuracy (61.12%-63.31%), attaining non-significant differences in the distribution of their accuracies across scenarios (Paired $t$-test, minimum Bonferroni corrected $p$-value of 0.444 across pairwise comparisons). For a fixed architecture, pretraining with increased image size helps improve accuracy somewhat on the OCP prediction task (SVG $128 \times 128$, rightmost periwinkle bar, vs. SVG $64 \times 64$, leftmost one), along with temporal augmentations for a high capacity model (FitVid, rightmost dark green bar vs. leftmost one), but these differences are overall non-significant (Paired $t$-test $p = 0.461$ and $p = 0.196$ within the SVG and FitVid architectures, respectively). However, not all end-to-end models match human accuracy well. In particular, we see that the object-slotted C-SWM class of models matches human accuracies *least* well compared to other model classes, despite varying the encoder size. The class of latent future prediction models overall better matches these behavioral patterns than the more explicitly structured C-SWM models, but there does not appear to be strong differentiation across foundation models, and also not much differentiation between having a dynamics module vs. using the encoder latents directly (rightmost two bars vs. leftmost bar in each group). This suggests that at least for the OCP task, either Physion is not sufficiently high-variation enough to pretrain the dynamics module or most of the predictivity comes from the physical scene understanding encapsulated in the foundation model encoder latents. The latter appears to be more likely, since pretraining the dynamics module on a larger-scale dataset such as Kinetics-700 for a subset of the models (VIP+LSTM/CTRNN, VC-1+LSTM/CTRNN, and R3M+LSTM/CTRNN; hatched bars) does not improve OCP accuracy over the base encoder latents either, relative to pretraining them on Physion (Paired $t$-test, minimum Bonferroni corrected $p$-value of 0.066 across architectures).

Additionally, we can look at finer-grained error patterns of the probabilities reported by humans compared to those of models (rather than accuracies alone), summarized in Figure 2B. Here we see that despite the metric being more detailed, humans are quite consistent with each other, suggesting that this type of behavioral metric is quite reliable (grey horizontal line). In fact, all models appear to be further from the human-to-human consistency than the OCP accuracy. However, overall, similar trends appear to hold across models as with the OCP accuracy measure – where the end-to-end pixel-wise models (FitVid and SVG) match these consistency scores the best across models, and the object-slotted C-SWM models match them the least well. From an AI perspective, it is actually quite relevant to work towards matching human error patterns, as it is highly correlated with the primarily performance based measure of OCP accuracy, as seen in Figure 2C ($R \approx 0.865, p \ll 0.001$).

And as shown in Figure 2D, all models have the most room to improve to match human error patterns and OCP accuracy (Figure S4) for soft-body interactions (the "Drape" scenario). On the other hand, models seem to do reasonably well on certain rigid-body scenarios, especially "Support" relations where stacks of objects fall over depending on their shapes and arrangement; "Collide", where pairs of objects collide depending on their trajectories and placement; and "Roll", where objects move across a surface either by rolling or sliding. In these scenarios, the best Physion-pretrained models, namely the pixel-wise future predictors, can attain human consistency scores much higher than what we report in Figures 2B and D of around 0.6 (cf. Figure S5B).

## 5 Comparison to Dense Neurophysiological Data

To gain more insight into model generalization, we compared the above models to neural dynamics recorded from macaque dorsomedial frontal cortex (DMFC), which was shown by Rajalingham et al. [2022b] to simulate the ball's position in Mental-Pong while behind an occluder, until it was intercepted by the paddle. This environment is completely out-of-distribution for the models, unlike Physion, and therefore tests structural generalization.

### 5.1 Inter-animal consistency and neural behavioral decoders

For each of the 79 conditions (different randomized start position of the ball), we present the frames in the visible epoch (the time up until the ball reaches the occluder) as context frames to the models, and unroll the model dynamics during the occluded epoch of the condition. We then build detailed, *physical* mappings (schematized in Figure 3A) from the committed model layer latent dynamics to match every single neuron in the population, and the ground truth ball state while occluded, when mental simulation takes place. We enforce the requirement, established in prior work [Nayebi et al., 2021, Cao and Yamins, 2021], that models are at least as similar to a given animal's neural responses as two conspecifics' neural responses are to each other. When we perform these mappings, we see that both the inter-animal neural predictivity consistency and ground truth ball state decoding from DMFC is quite high (grey horizontal lines in Figures 3B and C, respectively), indicating that these are very reliable neural and behavioral measures of the mental simulation phenomenon.

### 5.2 Neural response predictivity strongly separates models

When we map model units to DMFC units across 79 conditions (different randomized ball start position), we see in Figure 3B that across all architectures, only dynamically-equipped models from the video foundation model *class* predict DMFC dynamics best (46.34-48.83% neural predictivity), specifically in the latent space of the VC-1 and R3M encoders. However, the VC-1 and R3M encoder's latents alone are insufficient to predict the data (19.03% and 24.79% neural predictivity, respectively), although their latents best predict neural responses compared to other foundation models (cf. Figure S3C). Pretraining end-to-end on Physion with either a pixel-wise loss (SVG and FitVid; 14.44%-25.35% neural predictivity) or an object-slot loss (C-SWM with varying encoder sizes; 13.69%-23.05% neural predictivity) is not sufficient either, indicating a strong constraint on the neural mechanisms of mental simulation being performed on a suitable *latent* space. However, it is not the case that *any* latent space works, since all dynamics trained on encoder latents pretrained on *static* images, regardless of their imageset scale, attained at most 29.67% neural predictivity. This is suggestive that for a relatively small but time-varying stimulus such as the Mental-Pong ball, models that are pretrained on static images may not be equipped to handle the temporal coherence of a single object moving through time, as prior studies have mainly examined generalization of

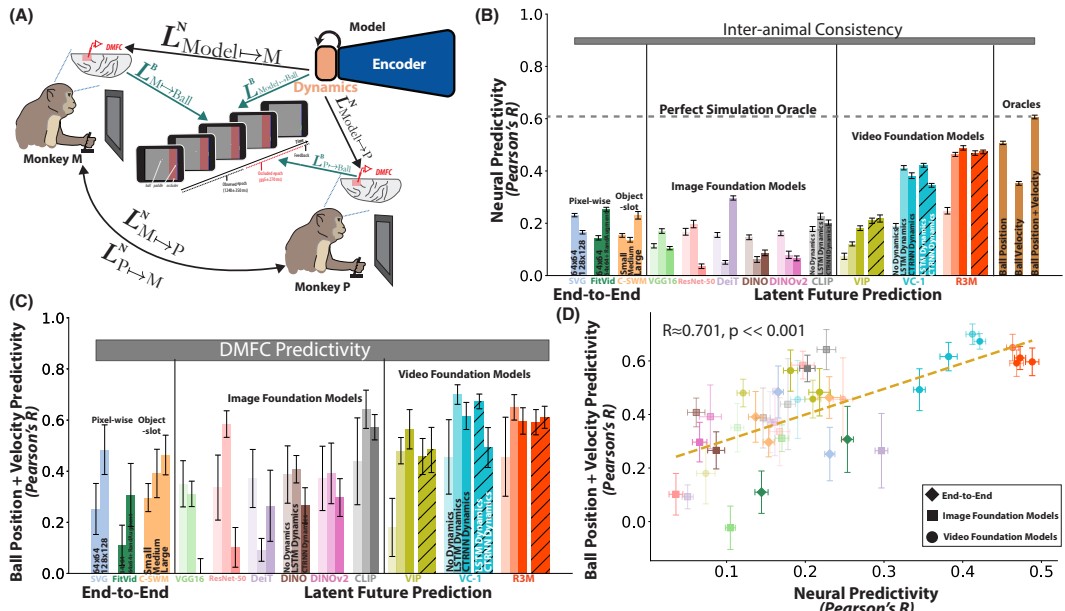

Figure 3: **Model Comparisons to Neural Response Dynamics:** **(A)** We measure how similar model dynamics (red) are to each primate's DMFC neural dynamics up to at least how (linearly) similar primates are to each other, via the linear transform $L^N$ (black arrows). We also assess how well models can decode ball state (position and velocity) up to how well it can be decoded from DMFC via the linear transform $L^B$ (teal arrows). **(B)** DMFC test set neural predictivity of each model, averaged across five train-test splits. Median and s.e.m. across 1,889 recorded units. For each of the latent future prediction models, there are three bars: the first corresponding to final context encoder latent itself ("No Dynamics"); and the last two bars are with LSTM or CTRNN dynamics, respectively. For the VIP, VC-1, and R3M based models, we also pretrain the LSTM and CTRNN dynamics on Kinetics-700 rather than Physion, and their predictivity is displayed in the last two additional hatched bars. The grey horizontal bar represents how well each primate's DMFC can predict the other's via the linear transform $L^N$ schematized in (A), setting a consistency ceiling on model predictivity. Position, velocity, and position + velocity oracles are the ground truth ball state values for those variables while it is occluded (and *not* visible to the primate during game play). Dotted "Perfect Simulation Oracle" line corresponds to the position + velocity oracle's neural predictivity. **(C)** Ball position and velocity predictivity of each model, averaged across five train-test splits, from the best DMFC fitting model layer used in (B) via the linear transform $L^B$ schematized in (A). The grey horizontal bar represents the ball position and velocity predictivity from DMFC units. Median and s.e.m. across four quantities: the ground truth $(x, y)$ position and velocity of the ball while it was occluded (in degrees of visual angle). **(D)** Mental-Pong ball position and velocity predictivity ($y$-axis) vs. neural predictivity ($x$-axis). Dotted golden line represents best linear fit to the data.

ImageNet-optimized CNNs to novel yet static, single-object scenes [Hong et al., 2016]. Moreover, even for video optimized foundation models, we see differentiation at the level of loss function. In particular, *goal-conditioned* self-supervised learning on videos, as instantiated by VIP (which differs from R3M primarily in the choice of objective function, as it shares the same ResNet-50 architecture and is also pretrained on Ego4D), fails to match DMFC neural dynamics well (7.37%-18.12% neural predictivity). We also see parsimony at the level of the architecture and pretraining dataset of the dynamics module, as the VC-1/R3M+CTRNN either attains comparable or improved neural predictivity over the more sophisticated VC-1/R3M+LSTM (middle vs. rightmost bars for each encoder type), even when pretraining on a larger dataset like Kinetics-700 (Figure S1). This suggests that simple dynamics may be sufficient at predicting neural response dynamics given the appropriate visual encoder and highlights the limits of dataset scale alone without changing inductive biases. Both the VC-1/R3M+LSTM and VC-1/R3M+CTRNN models approach the neural predictivity of the ground truth ball position oracle model (50.74%, leftmost golden bar in Figure 3B), with the joint position + velocity oracle performing the best (60.65% neural predictivity, rightmost golden

bar, which also corresponds to the "Perfect Simulation Oracle" dotted line). This suggests that a substantial amount of the neural response variability is devoted to simulating the ball's state while it is occluded, as opposed to other static features of the environment. The remaining gap between the "Perfect Simulation Oracle" and the inter-animal consistency may be due to other factors such as time since stimulus onset, attentional demands/eye movements, and perhaps uncertainty when the ball bounces, which can be explored in future work.

In Figure 3C, we see that the most neurally predictive dynamically-equipped VC-1/R3M-based models *generalize* to the Mental-Pong task, approaching DMFC's ability to track the ground truth position and velocity of the ball while it is occluded, despite not being explicitly pretrained in this environment. In particular, there is a linear relationship ($R \approx 0.701, p \ll 0.001$) between the model's ability to generalize to the Mental-Pong environment and its ability to predict DMFC neural dynamics (Figure 3D). This relationship indicates that predicting the underlying neural dynamics is in fact behaviorally relevant to effectively simulating the ball's dynamics, which Rajalingham et al. [2022a,b] judged in humans based on eye tracking data. Furthermore, the dynamically-equipped VC-1/R3M-based models can best represent the ball position and velocity *separately* as well (Figure S6), demonstrating that they can independently track these variables.

## 6 Dynamically-Equipped Video Foundation Models Can Match Both Human Behavioral and Neural Response Patterns *Across* Environments

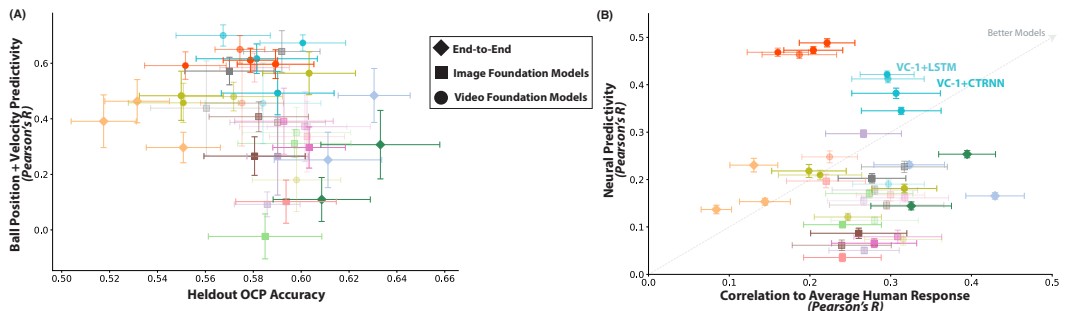

Figure 4: **Dynamically-Equipped Video Foundation Models Can Match Both Neural and Behavioral Metrics:** **(A)** Ball state (position and velocity) predictivity in Mental-Pong ($y$-axis) vs. OCP accuracy in Physion ($x$-axis). Across models, these two metrics appear to be largely independent. **(B)** The dynamically-equipped VC-1 models (VC-1+LSTM/CTRNN pretrained on either Physion or Kinetics-700) can reasonably match neural response dynamics in Mental-Pong and match human error patterns in the OCP task, relative to other models. Dotted grey line is the unity line to indicate that better models will occupy the top right of this scatter plot.

Both the OCP human behavioral error pattern metric and the DMFC Mental-Pong neural predictivity metric are linearly correlated to their corresponding behavior of OCP accuracy and ball position prediction (as seen in Figures 2C and 3D), which suggests that these more fine-grained metrics are relevant to the underlying behavioral goal that they represent. But how do they relate to each other, and is there a model class that can match *both* reasonably well?

As we can see in Figure 4A, the two behavioral goals of Mental-Pong ball state predictivity and Physion OCP accuracy do not appear to be very related across models ($R \approx -0.241, p \approx 0.134$). This suggests that held-out accuracy to novel scenes *within* environment as in OCP, does not imply that the same model will generalize to completely new environmental dynamics as in Mental-Pong. In particular, it is important to consider generalization to new environments, since the end-to-end pixel-wise models such as SVG and FitVid subtly overfit to this environment (attaining the highest OCP held-out scene accuracy), but they fail to generalize as well to the completely out-of-distribution Mental-Pong setting (rightmost periwinkle and dark green points in Figure 4A). For example, for FitVid, this failure is visualizable, as it predicts the Mental-Pong ball to be *static*, regardless of enlarging the ball during evaluation or pretraining with temporal augmentations (Figure S2).

Delving into the more challenging, fine-grained measures of DMFC Mental-Pong neural predictivity and correlation to human error patterns, we observe in particular that the dynamically-equipped

VC-1-based models (VC-1+CTRNN and VC-1+LSTM in cyan) reasonably match *both* human error patterns in Physion and Mental-Pong DMFC neural dynamics (Figure 4B).

Just as VC-1 on its own does not universally dominate on every individual sensorimotor task, but instead outperforms the best prior existing visual foundation models on average across the 17 CortexBench tasks [Majumdar et al., 2023], our finding that future prediction in the latent space of this model class reasonably matches both human behavioral patterns and neural responses is therefore consistent with this observation. Taken together, our results suggest that future prediction in the latent space of video foundation models for Embodied AI is a promising paradigm for developing models of physical dynamics that are both a better match to neural and behavioral recordings, and can structurally generalize across diverse environments and scenarios within.

## 7    Discussion

Overall, we find that structural generalization to completely new environments and matching dense neurophysiological data to be a strong constraint on sensory-cognitive hypotheses of mental simulation. In a manner evocative of Yamins et al. [2014]'s discovery of then-novel image categorization-optimized CNNs outperforming all prior models in matching primate inferotemporal cortex, our dynamically-equipped video foundation models notably outperform all other models in DMFC neural response matching, illustrating the evolving nature of goal-driven models, especially when applied to higher-cognitive brain areas while the animal is performing a task. Going forward, we believe that we are at a crucial turning point whereby foundation models that engage with the visually rich, dynamic scenes that humans and animals naturally interface with, will be jointly critical for progress in Embodied AI and neuroscience, addressing the recent call to action to build AI that can be as grounded in the physical world as animals are [Zador et al., 2023]. We observe that both the popular machine learning paradigm of pixel-wise future prediction and visual foundation models pretrained on webscale imagesets, typically favored for classic computer vision tasks like image segmentation and classification, underperform here. Instead, future prediction within the latent space of a video foundation model, pretrained on diverse egocentric sources, aligns best with high-throughput neural and behavioral patterns in scenes that it was not originally trained on. Our findings indicate that primate mental simulation harbors robust inductive biases, and is so far most consistent with predicting the future state of its environment in a latent space that is *reusable* across dynamic environments.

On our existing benchmarks, there are a few ways that we envision our current models can be improved. First, we believe that major strides can still be made in the encoder module of the models, namely by better leveraging temporal relationships to build a more "factorized" *and* reusable, object-centric representation. The need for a more factorized, object-centric representation is suggested by the high neural predictivity of the joint, ground truth position + velocity oracle in Figure 3B, compared to either the position or velocity oracles alone. At the same time, it is crucial to also maintain reusability in this representation since the fixed object-slot C-SWM models do not match these data well, and can be considered an example of an object-centric, but *not* reusable, representation. Therefore, a couple ideas in this direction would be to employ dynamic object-slots [Traub et al., 2022, Didolkar et al., 2023] or structured masking [Bear et al., 2023, Yuan et al., 2023] pretrained on Ego4D or even larger egocentric datasets such as CortexBench [Majumdar et al., 2023], to yield *object-centric, video foundation models*. Additionally, simpler modifications such as differentiating between agent-based egomotion vs. external object motion may likely improve video foundation models by better closing the loop between taking actions in the world vs. simulating it.

The encoder modifications mentioned above may enable the models to learn more explicit representations of objects and their material properties, compared to current self-supervised methods that are more statistical in nature [Balestriero et al., 2023]. This is especially relevant given our finding that all current models struggle most with the soft-body interactions of the "Drape" scenario in Physion, both in terms of accuracy (Figure S4D) and matching human error patterns (Figure 2), yet Ego4D has many examples of deformable objects in it (such as baking, clay, etc). Thus, it is *not* the pretraining dataset that is impoverished, but rather the current model architectures and loss functions are failing to pick up on these in existing datasets. The dynamics architecture could also benefit from including multiple timescales of hierarchy, with some recent neurobiological evidence of this type of temporal hierarchy already existing in frontal cortex [Sarafyazd and Jazayeri, 2019]. Such dynamics could sync well with dynamic object-slot encoders, capturing more rapid object state changes at one timescale and slower material property shifts across scenes at a higher-level, more abstracted timescale.

## 8 Acknowledgments

We thank the anonymous reviewers for their helpful feedback on the initial manuscript. A.N. thanks Christopher J. Cueva, Leslie P. Kaelbling, Thomas Serre, Kevin A. Smith, Joshua B. Tenenbaum, Rahul Venkatesh, Nicholas Watters, and Chengxu Zhuang for helpful discussions. A.N., M.J., and G.R.Y. acknowledge the generous support of the K. Lisa Yang Integrative Computational Neuroscience (ICoN) Center at MIT. M.J. is also supported by NIH (NIMH-MH122025), the Simons Foundation, the McKnight Foundation, and the McGovern Institute. R.R. is supported by the Helen Hay Whitney Foundation.

## Broader Impact

Almost everything humans and animals do revolves around an intuitive understanding of the physical dynamics of the world we are embedded in, enabling us to make long-range plans and perform actions that ensure our survival. Furthermore, engineering at least this same level of intuitive physical understanding *in silico* will be critical for progress in robotics, autonomous vehicles, and any other embodied applications that involve *safely* taking actions in the real world. Our work not only provides a strong measurement of the degree of alignment between our current best engineered systems to those of humans and animals, but through the differentiation across choices of the architecture, objective function, and pretraining dataset, also provides scientific insight into the evolutionary constraints underlying the neural mechanisms of mental simulation. In particular, we quantitatively observe that future prediction in a latent space optimized for diverse, dynamic scenes is our current best theory that predicts neural dynamics during mental simulation behavior, compared to popular pixel-wise and object-slot alternatives. This set of observations suggests that making progress in Embodied AI will also correspondingly yield an improved understanding of mental simulation in human and animal brains.

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
