## Limitations

While our study identifies clear separations between model hypothesis classes, our best models still have not reached the consistency ceiling of the neural and behavioral benchmarks we have compared against. The latent future prediction *dynamics* modules of all the foundation models were pretrained on Physion just as the end-to-end models were, and those Physion trained dynamics modules were evaluated against neural and behavioral data, ultimately outperforming the end-to-end Physion dynamics. Despite our interest, pretraining the end-to-end models on datasets larger than Physion exceeds our current computational resources, as evidenced by models like FitVid requiring nearly a month of training on eight A100 GPUs with Physion alone. Therefore, the vision foundation models ultimately have to deal with the harder problem of generalizing to Physion compared to end-to-end models. While we believe our dynamically-equipped foundation model paradigm to be a generally promising way forward towards models with strong internal simulations, we identify in the Discussion (§7), several ways that their encoder and dynamics modules can be improved, which we plan to explore in future work. Overall, we believe it will be important for foundation models to learn more factorized representations of object state dynamics, which we provide quantitative, neurophysiological evidence for through the testing of a variety of oracle models. Finally, as we saw, dense neurophysiological data was a strong constraint on the models. We therefore believe that neural recordings from animals performing mental simulations in richer, multi-object environments will further constrain models, allowing us to better pinpoint the neural mechanisms of this behavior.

## A  Model Pretraining

Model code and weights are available here: `https://github.com/anayebi/mental-sim`.

All models are supplied with $T = 7$ initial context frames, and are trained with the Adam optimizer [Kingma and Ba, 2015] using PyTorch 2.0 [Paszke et al., 2019]. For all models except FitVid and SVG $128 \times 128$ (SVG trained on $128 \times 128$ pixel images), a single NVIDIA A100 GPU was sufficient to train it. FitVid required 8 A100 GPUs, and SVG $128 \times 128$ required 2 A100 GPUs.

### A.1  Physion Dataset

All models were simultaneously trained across all eight scenarios of the Physion Dynamics Training Set, constituting around 16,000 total training scenarios (2,000 scenes per scenario) [Bear et al., 2021], with a temporal subsample factor of 6 frames for a total sequence length of 25 frames that is randomly sampled.

Pre-existing end-to-end models were trained with their previously prescribed hyperparameters, and we always ensured that their training loss converged on the Physion dataset. FitVid Babaeizadeh et al. [2021] was trained on its standard $64 \times 64$ pixel input (both with and without RandAugment [Cubuk et al., 2020]), with its recommended batch size of 128 with a learning rate of 1e-3 and gradient clipping of magnitude 100, for 3125 epochs. When applying RandAugment across frames, it was crucial to fix the random seed of the cropping operations, to ensure that the *same* random transformation was applied across *all* frames in a single sequence. SVG [Denton and Fergus, 2018] was trained on either its usual $64 \times 64$ or $128 \times 128$ pixel inputs, with a batch size of 100 with a learning rate of 2e-3 for 300 epochs, as recommended in the original paper. Each C-SWM [Kipf et al., 2020] model was trained on $224 \times 224$ pixel inputs, with $N = 10$ object slots and its recommended batch size of 32 with a learning rate of 5e-4 for 100 epochs.

Given $T$ context frames, all of the two-stage dynamics models (LSTM and CTRNN) are trained on top of a frozen pretrained foundation model, to predict the encoder latent at the next timestep $T + 1$ with a mean squared error loss function, using a batch size of 32 with a learning rate of 1e-4 (except for VGG16 and DeiT, which used a learning rate of 1e-2) for 100 epochs. All models were trained with $224 \times 224$ pixel inputs, using the same image augmentations used in the original evaluation of the corresponding foundation model. "No Dynamics" corresponds to fixing and propagating the last context latent of the foundation model through time, and therefore does not require any additional training.

### A.2  Kinetics-700 Dataset

We additionally trained the dynamics of the two-stage VIP+LSTM, VIP+CTRNN, VC-1+LSTM, VC-1+CTRNN, R3M+LSTM, and R3M+CTRNN models on Kinetics-700 [Carreira et al., 2019]. We first wrote out each frame of the mp4 videos as a jpeg with the smallest side set to 480 pixels via inter-area interpolation. We then trained the models on videos of length at least 8 or more frames (since we need at least $T = 7$ context frames and one additional frame to predict), that is randomly sampled during training. This constraint resulted in 536,640 training videos, and 33,966 validation videos. We trained models for 100 epochs each on a single A100 GPU with a batch size of 256 with a linearly rescaled learning rate of 8e-4 [Goyal et al., 2017].

# B  Comparison to Human Physical Judgements

Given $T = 7$ initial context frames, each model that we consider above is stimulus-computable and can be decomposed into an encoder $\mathcal{E}$ that yields a state representation per frame and a dynamics module $\mathcal{D}$ predicts unseen states given the prior state observations from the encoder. We first freeze model parameters and present all models with the eight Physion Readout Training Set scenarios ($\sim 1,000$ stimuli per scenario) for 25 timesteps with a subsample factor of 6 frames (same as during Dynamics Training in Section A.1). We combine the observed context and simulated model dynamics, which corresponds to the recommended *all observed+simulated* protocol in Physion, as it is agnostic to any particular scenario and best tests general physical understanding insofar as it can be assessed by Physion [Bear et al., 2021]. Specifically, we flatten these outputs and train a logistic regression classifier (cross-validated via a stratified 5-fold procedure) for 20,000 total training iterations to ensure convergence. To test the models after training the readout, we then fix the readout and present the same stimuli that the 100 human participants saw. For each stimulus, we compute the proportion of "hit" responses by humans, and correspondingly we will extract the hit probability generated by the logistic regression readout from the models. The **Correlation to Average Human Response** is the Pearson's correlation between the model probability-hit vector and the human proportion-hit vector, across stimuli *per* scenario. The **Heldout OCP Accuracy** of humans and models is the average accuracy, across stimuli *per* scenario.

To give the final values of the two quantities, we then compute the weighted mean and s.e.m. of the above per scenario quantities $x_i$ across scenarios, weighted by the number of stimuli $w_i$ per scenario[1], computed as such:

$$\text{weighted\_mean} = \frac{\sum_{i=1}^{s} w_i x_i}{\sum_{i=1}^{s} w_i}.$$

$$\text{variance} = \frac{\sum_{i=1}^{s} w_i (x_i - \text{weighted\_mean})^2}{\sum_{i=1}^{s} w_i},$$

$$\text{effective\_sample\_size} = \frac{\left(\sum_{i=1}^{s} w_i\right)^2}{\sum_{i=1}^{s} w_i^2},$$

$$\text{weighted\_sem} = \sqrt{\frac{\text{variance}}{\text{effective\_sample\_size}}},$$

where $s = 8$ is the number of scenarios.

# C  Comparison to Neurophysiological Recordings from Macaque Dorsomedial Frontal Cortex (DMFC)

To perform the model to brain (and brain to brain) comparisons, we first produce an $(N_{\text{cond}} N_{\text{frames}}) \times N_{\text{units}}$ matrix representation for both models and primate DMFC, following a similar procedure used by Rajalingham et al. [2022b]. For each of the $N_{\text{cond}} = 79$ conditions, we present the models with $T = 7$ uniformly sampled context frames from the Mental-Pong stimulus up until the last frame that the ball is visible. We then use the above determined temporal spacing from the context frames to determine the number of roll-out steps for the model up until the total number of frames for the current video ($89 \leq N_{\text{frames}} \leq 217$ frames across conditions). Note that these values are therefore different for each condition, but always the same across all models. Neural responses from dorsomedial frontal cortex (DMFC) are originally in 50*ms* bins, which we also interpolate to match the number of frames for the current video. Monkey $P$ had $1,552$ units recorded with 64-channel linear probes (Plexon V-probes) and monkey $M$ had 337 units recorded with high-density 384-channel silicon probes (Neuropixels), resulting in 1,889 recorded units in total.

Once model and neural responses were temporally upsampled to match the number of frames of each video, we then compared their dynamics in the timepoints when the Mental-Pong ball was occluded. This was done by training a single cross-validated Ridge regressor $L^N$ shared across timepoints and conditions that maximized the neural predictivity of responses between monkey $P$ and monkey $M$. The Ridge regressor was always refit to each source but using the *same* cross-validated hyperparameters found between animals, and was trained on 50% of the conditions (and their corresponding occluded epoch timepoints), and tested on the remaining 50% of conditions, across five train-test splits (cross-validation was done separately per train-test split via a grouped five-fold iterator). All neural predictivities are reported on *heldout* conditions and their timepoints. For each animal $A \in \{P, M\}$ and a given train-test split, heldout neural predictivity *per unit* in $A$ is measured on the test

---

[1]Specifically, there were 150 stimuli in Dominoes, 149 in Support, 150 in Collide, 150 in Contain, 150 in Drop, 150 in Link, 94 in Roll, and 149 in Drape.

set as:

$$\mathrm{NP}(S,A) := \frac{\mathrm{Corr}(L^N_{S \mapsto A}(S), A)}{\sqrt{\widetilde{\mathrm{Corr}}\left(L^N_{S^1_{1/2} \mapsto A^1_{1/2}}\left(S^1_{1/2}\right), L^N_{S^2_{1/2} \mapsto A^2_{1/2}}\left(S^2_{1/2}\right)\right) \cdot \widetilde{\mathrm{Corr}}\left(A^1_{1/2}, A^2_{1/2}\right)}}, \qquad (1)$$

where the source $S \in \{\text{Model}, P, M\}$; $A$ is the trial-averaged neural response of animal $A$; $A^1_{1/2}$ and $A^2_{1/2}$ are the trial-averaged responses to the first and second random split-half of responses in $A$, respectively; $S^1_{1/2}$ and $S^2_{1/2}$ are the trial-averaged responses to the first and second random split-half of responses in $S$, respectively (when $S$ is a model, then $S = S^1_{1/2} = S^2_{1/2}$); Corr is Pearson's correlation; and $\widetilde{\mathrm{Corr}}$ is Spearman-Brown correction applied to Pearson's correlation since the numerator uses the full set of trials and the denominator uses half the available trials. The rationale of using a reliability-adjusted correlation is to account for variance that arises from noise in neural responses that no model can be expected to predict, as it is not replicable by experiment condition (for a detailed derivation, refer to [Nayebi et al., 2021, Appendix §B.1]). This quantity is then averaged across the five train-test splits, and aggregated across units in both monkeys $P$ and $M$, and we report the median and s.e.m. across all 1,889 units in both monkeys. Note that if $S$ is a perfect replica of $A$, then this quantity will be 1.0, regardless of the finite amount of data collected.

When building neural behavioral decoders to the ball's $(x, y)$ position and velocity, we concatenated monkey $P$ and $M$'s neural responses during the occluded epoch, and regressed it against the interpolated ground truth ball position and velocity (measured in degrees of visual angle). This five-fold cross-validated Ridge regressor $L^B$ was optimized to maximize the median of these four values using the same measure in equation (1) with $A$ replaced by the ball state instead. The ball state (position + velocity) predictivity of each model is therefore the median and s.e.m. of these four values as measured by equation (1), with the regressor $L^N$ replaced by $L^B$.

# D   Supplementary Figures

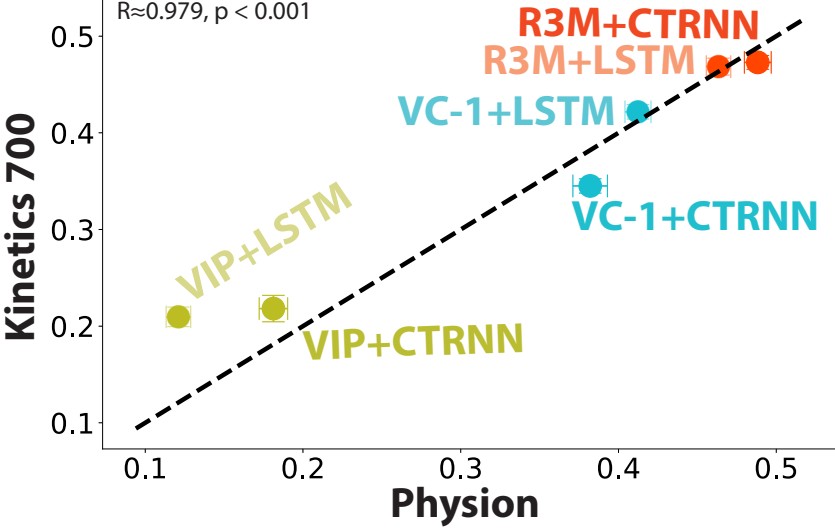

Figure S1: **Pretraining on Kinetics-700 Does Not Improve Neural Predictivity Over Pretraining on Physion:** Neural predictivity of VIP, VC-1, and R3M equipped with CTRNN or LSTM dynamics, which are pretrained on Physion ($x$-axis) vs. Kinetics-700 ($y$-axis). Dotted black line represents unity line.

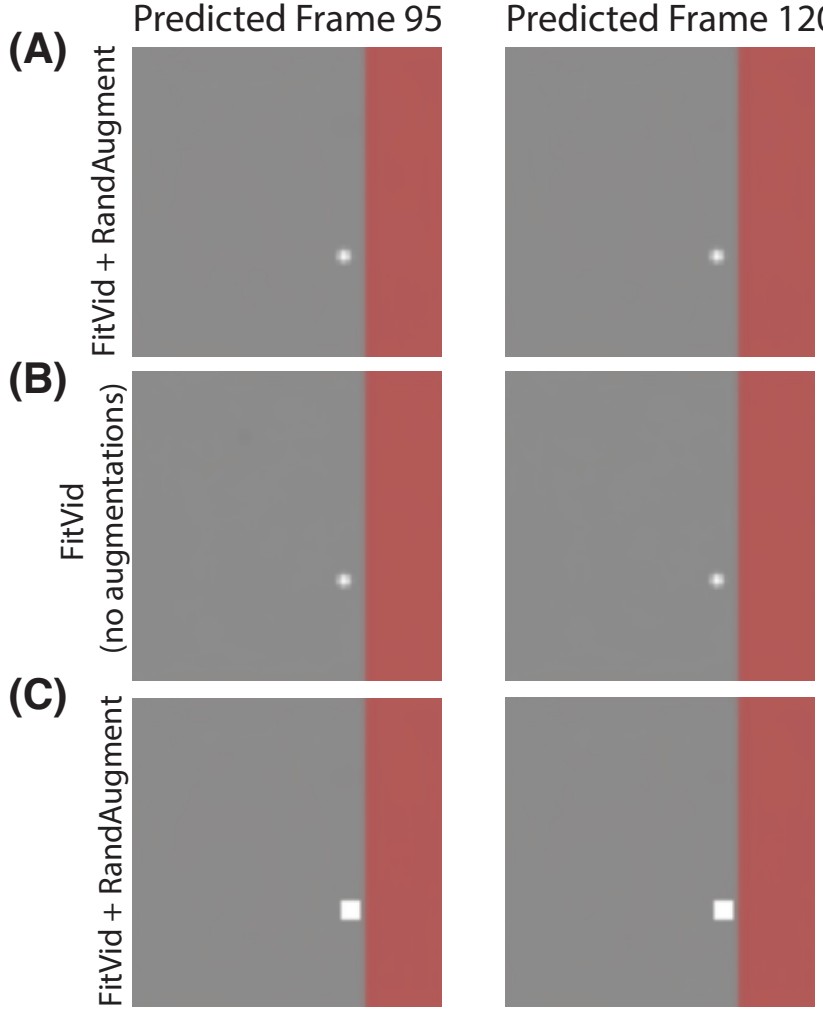

Figure S2: **FitVid Fails to Track Mental-Pong Ball:** Example predictions of FitVid across frames, when evaluated on a given Mental-Pong video, after being **(A)** pretrained on Physion with RandAugment, **(B)** pretrained without augmentations, and **(C)** pretrained with RandAugment but with the Mental-Pong ball enlarged to be a square during evaluation.

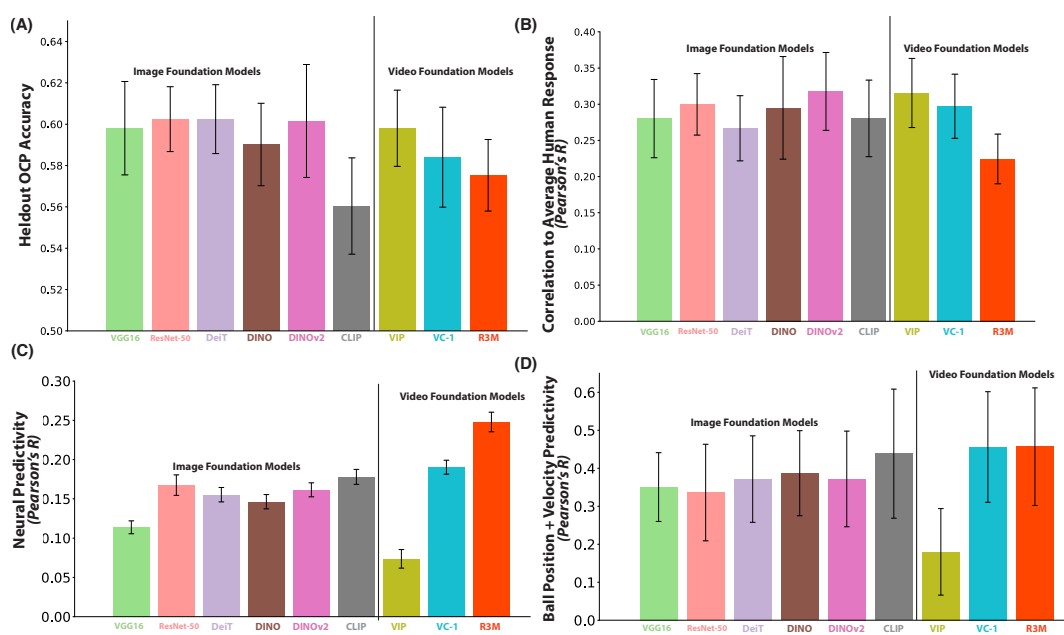

Figure S3: **Predictivity of Foundation Model Latents:** Evaluations of the fixed latent representations (no dynamics) of the nine classes of foundation models on **(A)** accuracy on the OCP task (cf. Figure 2A), **(B)** human judgement probabilities on the OCP task (cf. Figure 3B), **(C)** DMFC neural predictivity on Mental-Pong (cf. Figure 3B), and **(D)** Mental-Pong ball position and velocity predictivity (cf. Figure 3C). These were originally included in Figures 2-4.

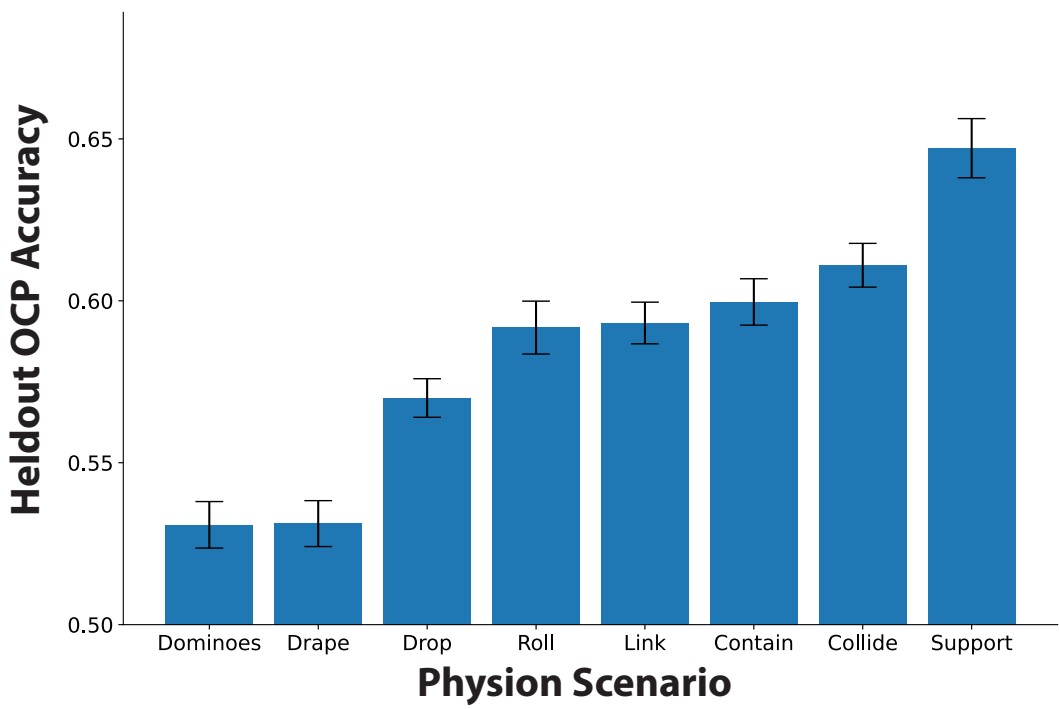

Figure S4: **Per Scenario OCP Accuracy:** Per scenario model comparisons to binary ("yes/no") accuracy on the OCP task (chance is 50%). Mean and s.e.m. across all 41 models.

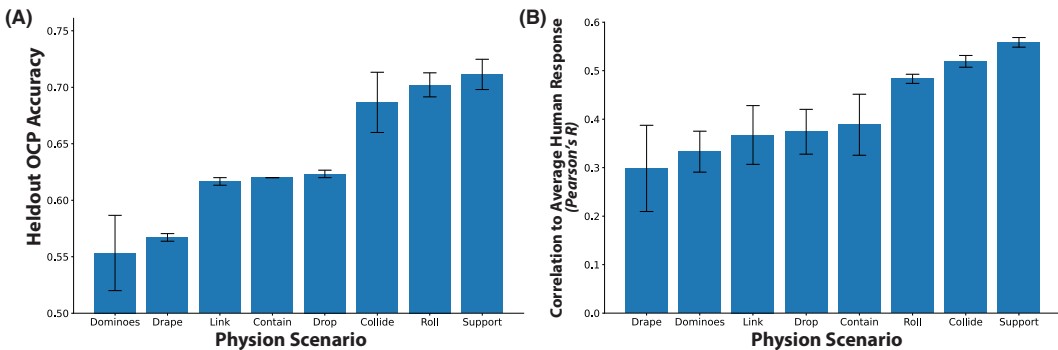

Figure S5: **Per Scenario OCP Evaluation of Best Pixel-Wise Future Predictors: (A)** Per scenario model comparisons to binary ("yes/no") accuracy on the OCP task (chance is 50%). **(B)** Per scenario model comparisons to human subject judgement probabilities for the OCP task. Mean and s.e.m. across the two best pixel-wise future predictors: FitVid pretrained on Physion with RandAugment and SVG pretrained on Physion with $128 \times 128$ images. Demonstrates that there are some models which match accuracy and human judgement probabilities on the OCP task quite well for certain scenarios (e.g. "Support", "Collide", and "Roll"), compared to the aggregate in Figure 2.

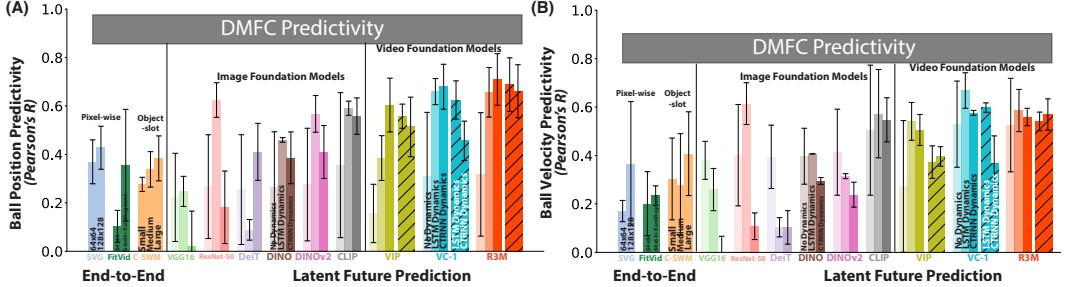

Figure S6: **Models Can Separately Represent Mental-Pong Ball Position and Velocity:** Same as Figure 3C, but separate predictivity of the **(A)** ball position and **(B)** velocity predictivity of each model, averaged across five train-test splits, from the best DMFC fitting model layer used in Figure 3B via the linear transform $L^B$ schematized in Figure 3A. The grey horizontal bar represents the ball position in **(A)** and velocity predictivity in **(B)** from DMFC units. Median and s.e.m. across two quantities: the ground truth $(x, y)$ position in **(A)** and velocity in **(B)** of the ball while it was occluded (in degrees of visual angle).