# OpenReview forum: "Neural Foundations of Mental Simulation: Future Prediction of Latent Representations on Dynamic Scenes"
_NeurIPS.cc/2023/Conference — NeurIPS 2023 spotlight_

### Official Review · Reviewer_bK8w · 2023-07-06

**Soundness:** 3 good
**Presentation:** 3 good
**Contribution:** 3 good
**Rating:** 7
**Confidence:** 2

**Summary:**

It has been hypothesized that the brain builds an internal model of its environment, and uses this model to make inferences and plan actions. This works aims to understand the neural mechanisms that are the basis of such computations. To this end, the authors construct several classes of artificial neural network models, each with different inductive biases, and evaluate them on future state prediction tasks in ethologically relevant environments.

They evaluate each of these models in two ways:

- comparison to human behavior in an object contact prediction task
- comparison to neurophysiological data from macaques playing mental-pong, a ball interception task in which the ball is partially occluded

From these evaluations, they identify that dynamics modules that predict the future state in the latent space of video foundation models reasonably matches both human behavioral patterns and neural responses across diverse environments.

**Strengths:**

Understanding the neural basis of mental simulation is an important problem both in neuroscience and AI.
The approach taken here is to construct several classes of artificial networks and evaluating them on tasks that require prediction of the future state of the environment. This allows the authors to probe which inductive biases are crucial for mental simulation.
The presentation is clear as well.

**Weaknesses:**

Not really a weakness, but details about the best fitting model layers would be useful to add. It would also be interesting to see some analysis comparing the best fitting latent layers across different models.

minor:
- line 17, 18 in the abstract talks about the mental pong task without introducing it

**Questions:**

- The R3M and VC-1 video foundation models (+ dynamics) are the best at neural response predictivity.  However, one is a ResNet and the latter is a transformer based model. Also the training objectives are different for both models. Why is the performance of these two models similar? Is it the scale and/or type of pre-training dataset?
- For the object-centric models did you try using recurrent networks for the dynamics module?
- Was any analysis performed to compare the best-fitting latent representations across models?

**Limitations:**

Limitations are addressed in the appendix.

---

> ### Author Rebuttal · Authors · 2023-08-08
>
> - *Not really a weakness, but details about the best fitting model layers would be useful to add. It would also be interesting to see some analysis comparing the best fitting latent layers across different models.*
>
> Absolutely agree. We actually compared the latent layers in Figures 2-4 across different models (these are the leftmost bars in each group of the nine image and foundation models classes that we tried). We found that the best latents overall were from the video foundation models (VC-1 and R3M) trained in a self-supervised manner on the Ego4D dataset. The dynamically-equipped versions of these models, optimized to predict the future state of the environment in this latent space, were the best across the neural and behavioral metrics we compared (Figure 4B). For clarity, we will include a plot with just the latent layers alone in Figure S3 of the revised manuscript. Thank you for the suggestion.
>
> - *line 17, 18 in the abstract talks about the mental pong task without introducing it*
>
> Thank you for pointing this out, we will introduce the Mental Pong task in the revised abstract prior to lines 17-18.
>
> - *The R3M and VC-1 video foundation models (+ dynamics) are the best at neural response predictivity. However, one is a ResNet and the latter is a transformer based model. Also the training objectives are different for both models. Why is the performance of these two models similar? Is it the scale and/or type of pre-training dataset?*
>
> This is a great question. We believe it is likely a combination of the self-supervised loss function applied to the Ego4D dataset. For example, both R3M and VIP have the same architecture (ResNet-50) trained on Ego4D, but VIP is trained using an offline RL (goal-conditioned) objective predicts neural responses poorly (Figure 3B, dark green bars) relative to R3M (even at the level of latents), whereas R3M is trained with a contrastive, self-supervised objective. Furthermore, when trained with a categorization objective on ImageNet, ResNet-50 performs poorly (Figure 3B, red bars, “Image Foundation Models”), even at the level of the latents (Figure 3B, leftmost red bar, “Image Foundation Models”). Similarly, though somewhat less directly, Transformers like DeIT, DINO, DINOv2, and CLIP are trained with self-supervised objectives, including contrastive ones, on webscale datasets even larger than ImageNet (“Image Foundation Models” bars in Figure 3B), yet perform worse than the self-supervised VC-1 transformer trained on Ego4D. We will include discussion of these points above in Section 5.2 of the revised manuscript. Thank you for bringing it up.
>
> - *For the object-centric models did you try using recurrent networks for the dynamics module?*
>
> Yes, the object-slot models are trained with a recurrent graph neural network (GNN) dynamics module, originally proposed by Kipf et al., 2020. We apologize for this not being clear in the text, and we will mention this more clearly in Section 3 when the model is first introduced. Thank you for asking this.
>
> - *Was any analysis performed to compare the best-fitting latent representations across models?*
>
> Yes, this is a great question. We actually evaluated the latent layers against our neural and behavioral metrics in Figures 2-4 (refer to the leftmost bars in each group of nine foundation models). The most effective latent representations were from the video foundation models self-supervised on Ego4D (VC-1 and R3M), and were most differentiated on neural responses (new Figure S3, panel C). Their dynamically-equipped versions, optimized to predict the future state of the environment in this latent space, excelled in all neural and behavioral metrics (Figure 4B). We will add a dedicated plot of the latent layers alone in a new Figure S3 in our revised manuscript for better understanding. We appreciate the recommendation.

---

> > ### Comment · Reviewer_bK8w · 2023-08-16
> > **response to rebuttal**
> >
> > Thank you to the authors for these responses. My questions have largely been addressed and I am happy to keep my rating.

---

### Official Review · Reviewer_tmZS · 2023-07-07

**Soundness:** 3 good
**Presentation:** 3 good
**Contribution:** 2 fair
**Rating:** 5
**Confidence:** 4

**Summary:**

The authors compare "foundation models" of vision for mental simulation. They consider several large models including models trained on static scenes and dynamic scenes. It was found that the models optimized with self-supervision on dynamic scenes yielded the best neural predictivity.

**Strengths:**

The work is interesting since mental simulation is relatively understudied but important for embodied models. The paper is clearly written.

* The work is thorough in testing out a lot of models and model families.
* Results on neural predictivity is strong for the video foundation models.

**Weaknesses:**

* Since all the video foundation models are trained with Ego4D and the image foundation models are trained with ImageNet, it is not straightforward to disentangle the effect of dataset vs role of dynamics itself.

* The models only differ in terms of neural predictivity while being similar to each other in accuracy and correlation to human responses. Previous work (Rajalingham et al. [2022b]) seem to support that model behavior should be measured in terms of correlation to human responses to disambiguate between models - with a high correlation suggesting that the model is using the same strategy as humans. In light of this, the results presented seem weaker than as claimed.

* I do not see the contributions of the paper beyond "large models trained with dynamic scenes data are better at neural predictivity for a dynamic task" which is not surprising.

**Questions:**

*Fixed encoder that is not trained during "dynamics" training seems too constraining (also unlike human brains). This might have unequal effects on the different models. For example, it might have hurt imagenet models more than others etc. Have you tried versions where you did update the encoders too?

*There seem to be some models (Fig. 3 D) that are very good in accuracy but not so good in neural predictivity. Do you know why this is happening?

---

> ### Author Rebuttal · Authors · 2023-08-08
>
> - *Since all the video foundation models are trained with Ego4D and the image foundation models are trained with ImageNet, it is not straightforward to disentangle the effect of dataset vs role of dynamics itself.*
>
> We wanted to understand this question too, and as a result, we included the latents (without any dynamics) in Figures 2-4 (refer to the leftmost bars in each group). Across all nine classes of image and video foundation models that we tried, the most effective latent layers were from the self-supervised video foundation models trained on Ego4D (VC-1 and R3M), as can be seen in Figure 3. However, the latent representations of VC-1 and R3M alone are significantly worse at predicting neural responses than adding dynamics on top of it (remaining bars in each group in Figure 3B). Furthermore, the dynamically-equipped versions of VC-1 excelled in all neural and behavioral metrics (Figure 4B) relative to their latents and those of other models. Thank you for raising this important point.
>
> - *Previous work (Rajalingham et al. [2022b])…support that model behavior should be measured in terms of correlation to human responses… In light of this, the results presented seem weaker than as claimed.*
>
> Rajalingham et al. 2022b trains and evaluates models in the same Mental-Pong environment; however, we only (*evaluate* our models in this environment since we are after the more challenging problem of task generalization to multiple diverse scenarios (in the Physion and Mental-Pong environments). For example, as we show in Figures 3C and S2, even state-of-the-art models like FitVid (a pixel-wise future predictor) fail to generalize to Pong after being trained in Physion, unlike our dynamically-equipped video foundation models. The reason why we focus on generalization to Mental-Pong is that monkeys can perform these tasks without substantial training, suggesting that they are already equipped with the necessary neural foundations for mental simulation in this environment. Therefore, we aim to also build networks that are not explicitly trained on Mental-Pong itself, but are tasked to generalize to this novel setting as a test of their general understanding of physical scene dynamics -- chiefly developed through three factors: their architecture, optimization objective, and pretraining on a naturalistic environment.
>
> - *I do not see the contributions of the paper beyond "large models trained with dynamic scenes data are better at neural predictivity for a dynamic task" which is not surprising.*
>
> Actually, not all large models trained on videos predict neural data well. In fact, a major point of our paper is that many state of the art machine learning models for future prediction fail to match neural and behavioral data well. For example, of the models trained on dynamic scenes, neither the pixel-wise future predictors nor the object-slot models do this well (“End-to-End models in Figure 3B), nor does the video foundation model VIP or its dynamically equipped version do it well either (green bars in Figure 3B). Our work therefore suggests strong constraints on models, specifically *not* through RL, pixel-wise losses, or bespoke object-slots, but rather through self-supervised pretraining from egocentric views that humans and animals naturally receive. Crucially, doing future prediction on this *reusable* latent representation is important (since the fixed latent alone is not sufficient, see the leftmost bars in Figure 3B in each group).
>
> - *Fixed encoder that is not trained during "dynamics" training seems too constraining (also unlike human brains)....Have you tried versions where you did update the encoders too?*
>
> That’s a great question, and one that we have looked into. In fact, our decision to use a fixed encoder during “dynamics” training, despite seeming restrictive, was largely influenced by our experimental findings. For example, prior state-of-the-art models like end-to-end pixel-wise or object-slot predictors did not yield superior results (cf. Figure 3 “End-to-End” bars) and struggled to generalize to Mental-Pong (cf. Figure 3C dark green FitVid bar, and Figure S2), even though their encoders were updated during training. Based on these observations, we saw that generalization to novel environments failed even when the encoders are allowed to update during training, especially if inductive biases such as the loss function and pretraining dataset are not properly chosen. In fact, this approach aligns with insights from more cognitively-inspired graph neural networks (GNNs), which propose to solve intuitive physics tasks by modeling dynamics on meaningful entities, like objects, their material properties, and their relations, rather than raw pixels. Since GNNs can match the OCP human error patterns well (as shown in Bear et al., 2021), they suggest that one can “factorize” the problem of intuitive physics into solving a challenging vision problem (e.g. an understanding of the scene) with dynamics that operates on that representation. These observations together motivated our fixed foundation model encoder with dynamics approach.
>
> Going forward, resources permitting, we are planning to explore hybrid models that train the encoder with a masked autoencoding (MAE) style objective applied to next-frame prediction on Ego4D and larger datasets like CortexBench. We believe these methods could better leverage temporal relationships to learn reusable *and* object-based latent representations.
>
> - *There seem to be some models (Fig. 3 D) that are very good in accuracy but not so good in neural predictivity. Do you know why this is happening?*
>
> Ball position & velocity offer coarser measures than neural predictivity, with far fewer tracked quantities (4 vs. 1889). Thus, many inductive biases are able to track 4 quantities, but far less can track 1889. Our goal is models that excel in **both** metrics, as shown with the dynamically-equipped *video* foundation models in Figure 3D (top right circles).

---

> > ### Comment · Reviewer_tmZS · 2023-08-17
> >
> > Thank you for the detailed response!
> >
> > > Rajalingham et al. 2022b trains ............... chiefly developed through three factors: their architecture, optimization objective, and pretraining on a naturalistic environment.
> >
> > To clarify my question - I am questioning why the results from Fig 2 B do not support your claims that video foundation models are better - the image models are as good. Rajalingham et al. 2022b supports that correlation to human response is what we should look at to adjudicate between models but here the results seems to suggest no superiority of the video models. Am I right in saying this?
> >
> >
> > I am cognizant of the breadth of this work but I was just expecting stronger results. I do see however that for neural data predictivity, the video models are better. I think many of my earlier reservations were addressed and I am compelled to increase my score.

---

> > > ### Author Response · Authors · 2023-08-17
> > > **Reply to Reviewer tmZS**
> > >
> > > Thank you for your consideration of our comments, and for your engagement, we appreciate it!
> > >
> > > - *I am questioning why the results from Fig 2 B do not support your claims that video foundation models are better - the image models are as good. Rajalingham et al. 2022b supports that correlation to human response is what we should look at to adjudicate between models but here the results seems to suggest no superiority of the video models. Am I right in saying this?*
> > >
> > > The behavioral data that Rajalingham et al. 2022a,b refers to is tracking of the Pong ball's position, which is judged in humans based on eye tracking data. Therefore, the most relevant data is Figures 3C & D, rather than Figure 2B -- we will make this connection clearer in the revision, thank you for mentioning it. These figures evaluate how well the models track the ball in the Pong environment. As you can see in Figure 3C, the video foundation models that best match DMFC neural dynamics (from Figure 3B) while the macaque plays Pong also best track the ball, approaching DMFC's ability to do so (rightmost red and blue bars approaching the grey horizontal line in Figure 3C). This correspondence between predicting neural response dynamics and tracking the ball is quantified in Figure 3D, and you can see the video foundation models do those both best (top right circles in Figure 3D).
> > >
> > > It is also worth noting that our human behavioral responses in Figure 2 are a completely different set of judgements from tracking the Pong ball. Specifically, these are object contact predictions in naturalistic 3D environments. Our main point with Figure 2B is to show that the pixel-wise models subtley "overfit" to the environmental statistics in which they are trained, and fail to generalize to the novel environment of Pong -- a hallmark of both human and macaque intuitive physics, as they can learn the Pong task almost immediately. In other words, the OCP task in Figure 2 involves held-out videos but in the *same* training environment as the models. Whereas, in Figure 3, Pong is a completely new environment, and we see that pixel-wise models (especially FitVid, dark green bars in Figure 3C) fail to track the ball. This failure is visualized in Figure S2, where the ball is held fixed in FitVid's simulation. Furthermore, in part due to the novelty of the Pong environment relative to the models and also having to track neural *dynamics* rather than a static yes/no judgement as in OCP, the neural data more strongly differentiates individual models than the human behavioral responses do (Figure 3B vs. Figure 2B). However, despite this, here again video foundation models (specifically the dynamically-equipped VC-1 models) match *both* human error patterns on OCP and neural responses in Pong reasonably well compared to all other models (shown in the blue circles in Figure 4B).

---

### Official Review · Reviewer_6vWo · 2023-07-09

**Soundness:** 4 excellent
**Presentation:** 3 good
**Contribution:** 4 excellent
**Rating:** 8
**Confidence:** 4

**Summary:**

The paper compares a rather large variety of DL models that are able to “future predict” environmental states, including pixel-based deep networks, compositional approaches (e.g., slot-wise processing objects), as well as image and video foundation models. In the latter case, the latent space of the foundation model is used to future predict. The models are then evaluated by comparing their prediction accuracy in an object contact prediction (OCP) task with human performance and in a mental pong task with macaque neuro-biological data.

**Strengths:**

The paper offers a huge data study that reveals model differences when predicting human behavioral performance and macaque neural dynamics data. At the moment, video-based foundation models yield best (but not very good and not much better compared to other models) performance. Generally, the generated data is interesting and should not be lost.

**Weaknesses:**

On the negative side, the paper focuses on only two tasks, which are not very well-motivated. I have found neither a clear motivation to choose these tasks nor a clear hypothesis what the authors would expect.

Further, the results that have been achieved are very far from human-to-human consistency / human accuracy and also very far from inter-animal consistency. This implies to me that the results and the performance differences between the models are not really due to actual systematic differences between the models and the model relations to how humans process the data; rather, they may be due to more fundamental design choices. Moreover, particularly in the OCP task, the model performance differences nearly seem random.

Surprising to me is the fact that object-centric models yield worst performance (but all are bad), although the task itself is suitable for an object-centric setup. Maybe the authors did not tap onto the appropriate latent state representation – or it this the case because they do not allow to actually simulate dynamics with the object-centric setup?

This concern goes hand-in-hand with the fact that the authors do not report an oracle-like performance in the OCP task (as they do in the mental-Pong dataset). From the CogSci work of Tenenbaum, Goodman and others it is well-known that humans do use an internal simulator to approximately solve such tasks… none of the systems the authors consider are able to actually run an internal simulation of the potentially unfolding dynamics. Rather, the authors seem to tap directly into the static latent state of the respective models.

In the mental-pong task, the authors do provide an oracle comparison, which indeed yields the best neural predictivity performance, followed by VC-1 and R3M, both of which can learn to predict ball position and velocity because they learn to predict video dynamics where entities, for example, fall, fly, and roll around. Thus, I do not really see any insight created by these comparisons, except for that a model is needed that predicts ball positions and velocities… in the end, the oracle should be the baseline, which needs to be beaten by a neural network to come closer to understanding how our brain / macaque brains solve this task.

Seeing this huge study, I am also quite concerned about the compute time (energy) invested here. The authors do not provide enough information about training etc. Thus, it is impossible to even estimate this investment.

The final discussion stays rather superficial on the actual insights gained (because unfortunately there are no significant insights really). The mere task to match human data / macaque data even better may be interesting for some sub-community along the neuro-DL interface, but I miss both cognitive and structural insights that go beyond the fact that object dynamics (including velocities) as well as object constellations need to be encoded in the considered tasks.


**Questions:**

A few more detailed comments:

The abstract starts very superficial – it took me a while to understand what you are actually after. A full re-write seems necessary.

In the introduction, the second paragraph emphasizes that “predicting the physical dynamics of environments” is critical… but isn’t this precisely what you emphasize in paragraph 1 of the introduction, that is, that mental models are important? Only mental (intuitive physics-approximating) models allow to predict physical dynamics, isn’t this per definition the case?

Over-general implications are drawn in the introduction – the results are not strong enough in my opinion to come to the put-forward conclusions. Moreover, the written implications stay on a rather superficial level, although the restuls are very particular to the considered two tasks and the methods employed.

Finally, on the methods side, more details would have been useful to compare sizes and understand the exact training procedure to fit the data.


**Limitations:**

not addressed

---

> ### Author Rebuttal · Authors · 2023-08-08
>
> - *…the paper focuses on only two tasks, which are not very well-motivated.*
>
> We thank the reviewer for suggesting the need for this clarification, and will add it to the Introduction. Specifically, the OCP task (Bear et al. 2021) tests realistic simulations of a wide range of everyday physical phenomena, including rigid and soft-body collisions, stable multi-object configurations, rolling, sliding, and projectile motion, in a realistic 3D environment with 2,000 scenarios per condition, and is therefore more comprehensive than previous behavioral benchmarks so far. Furthermore, Mental Pong is the *first neural* dataset that was shown to involve mental simulation, and is a dense neurophysiological dataset containing almost 2,000 neurons recorded with high temporal precision (Rajalingham et al. 2022b). They are therefore both high-throughput in behavior as OCP is (from 100 human subjects across 16,000 scenarios), or in the neural readout. In addition to these tasks, we benchmark a large variety of models (41 models total), none of which are strawman models – they can functionally perform the task on large-scale datasets and many of them are considered state-of-the-art. Yet, when compared our human behavioral and neural benchmarks, in part due to the high-throughput nature of our benchmarks, they strongly differentiate these functionally reasonable hypotheses.
>
> - *Further, the results that have been achieved are very far from…consistency*
>
> This is actually a major point of our work. Our work does not claim to have solved intuitive physics, but first shows the limitations of prevalent approaches and offers promising future directions validated against strong neural and behavioral benchmarks. Even though we utilize state-of-the-art machine learning models, they still significantly lag behind primate intuitive physics. As seen in Figures 2C and 3D, aligning with human behavior and neural dynamics directly pertains to solving the AI task in each environment. Notably, the widely-used pixel-wise future predictor excels in familiar settings but overfits in novel ones, like the Pong scenario. This underscores our emphasis on out-of-distribution structural generalizations, rather than the typical ML generalization where training and testing environments are statistically similar.
>
> Moreover, our work *also* points a concrete way forward to better models. In particular, we find that latent future prediction appears to be the most promising paradigm overall, compared to popular alternatives of pixel-wise future prediction, object slots, and the fixed latents alone of foundation models. In other words, the problem of intuitive physics can effectively be “factorized” into a challenging vision problem and then equipping that scene representation with dynamics. And perhaps more crucially, this visual representation needs to be reusable across dynamic scenes. This suggests that reusability is an important factor that future models should incorporate, and points to a move away from the commonly used web-scale imagesets (no matter their scale) that take snapshots with lighting conditions that tend to be unrealistic from the standpoint of egocentric viewpoints humans and animals receive. Furthermore, the dynamics architecture could benefit from better leveraging a more object-based representation of temporally-active state variables (as suggested by the high neural predictivity of the joint, ground truth position + velocity oracle; rightmost bar in Figure 3B). Taken together, these observations suggest that working on self-supervised loss functions that better leverage temporal relationships to learn object-based, video foundation models are a next step, coupled with latent future predicting dynamics on top of these reusable, object-based latent representations. One way toward that goal can be adapting the masked objective in MAE to operate on the future frame rather than the current frame, and train on a more diverse video dataset similar to CortexBench, rather than Ego4D alone.
>
> - *do they not allow to actually simulate dynamics with the object-centric setup?*
>
> Our object-slot models use a learned graph-neural network module on top to simulate the forward dynamics on top of the object-slot latent representation.
>
> - *the authors do not report an oracle-like performance in the OCP task (as they do in the mental-Pong dataset).*
>
> The CogSci work, as we discussed in Section 2, provides a near-oracle model for the OCP task using ground-truth simulator states, unlike real-world sensory inputs. We will indicate their numerical performance in the revision. With ground-truth inputs that these CogSci models take in, such as in Mental-Pong, the dynamics prediction problem becomes trivial since there is a single object, so our focus is on the much more challenging problem of bridging the gap from raw sensory inputs to behavior.
>
> - *none of the systems the authors consider are able to actually run an internal simulation of the potentially unfolding dynamics.*
>
> Actually, the dynamics are additionally trained on top of the latent representation, even in the cases when the visual encoder is fixed. We in fact train multiple dynamics architectures (CTRNN and LSTM) across all foundation models, on both Physion and the larger Kinetics 700 dataset.
>
> - *In the mental-pong task… I do not really see any insight created by these comparisons, since the oracle should be the baseline.*
>
> The oracle models are not baselines since they possess perfect information inaccessible to animals or pixel-based models, as the ball is occluded. Oracles reveal that using more object-centric priors, which access both ball position & velocity, outperforms relying on just one of these. Taken with Fig. 4B, these indicate that future models should learn reusable *and* object-based representations from egocentric videos.
>
> - *Limitations: not addressed*
>
> The Limitations section was in the initial submission, in the first paragraph of Supplementary Material.

---

> > ### Comment · Reviewer_6vWo · 2023-08-11
> > **still many open questions - but important things have been clarified - thank you!**
> >
> > From the general rebuttal reply by the authors, I am now much more convinced that this paper actually does have some merit, which I have overlooked in my review. I think what put me off was the huge analysis and the lack of a clear hypothesis. As the authors write in their general rebuttal, I highly recommend expanding on these aspects – particularly in Section 2. Additionally, the abstract and intro can be improved. I recommend getting to the point faster – specify exactly which data you are considering in your modeling work (behavioral and neural data from humans and primates). Then put forward your hypothesis and/or the main conclusions that you draw.
> >
> > Thereby, I am still confused why the slot-based models do not work well at all while you are suggesting that more “factorized representations of object state dynamics” will be important. Shouldn’t slot-based models produce such factorized representations?
> >
> > I also recommend not to emphasize the size of your data or the throughput overly much, but rather the intention you have in using it. In fact, could you explain to me (and probably many readers, I also asked some colleagues) what you actually mean by “dense” neurophysiological data? You highlight it also, for example, in the limitations and the abstract as an important factor to constrain your models. Similar, I am honestly not quite sure what you mean with “high-throughput human behavioral readouts”.
> >
> > Please also note that the need for out-of-distribution structural generalization is well-known in cognitive modeling (e.g., Lewandowsky, S., & Farrell, S. (2011). Computational modeling in cognition: Principles and practice. Sage Publications).
> >
> > I highly recommend discussing the large gap between human-to-human consistency and model-to-human consistency further --- this was one of my main concerns and still makes me hesitate to fully recommend publication. Are the models any good at all? Are model differences in terms of model-to-human correlation actually meaningful? I am still not quite sure.
> >
> > Please excuse that I had overlooked the limitation section in the appendix. I still find it rather short, though. With respect to the rather weak model comparison levels reaches, maybe a comparison to similar modeling approaches – say on the level of primary visual areas – would help? That is, a comparison about the levels such models are reaching and the level that you are reaching (compared to the human-to-human consistency)? The gap could tell a story about how much is missing.
> >
> > Thank you for clarifying the oracle value in the pong task – the ball is not visible behind the occluder – all considered models fail to track the ball, correct? Maybe a reference to the fact that babies are able to track the ball at least with 3 months of age if not earlier would be warranted. --- which again somewhat points out to me that the study is to a certain extent an overkill.
> >
> > Nonetheless, I fully agree to the insights and to the messages that you want to convey, which you have made now much clearer in the rebuttal. Seeing that comparisons like the one presented are en vogue, this work and the insights are timely and will gain recognition (and citations).
> >
> > I thus increase my score to weak accept.

---

> > > ### Author Response · Authors · 2023-08-14
> > > **Reply Part 1 of 2 to Reviewer 6vWo Comment**
> > >
> > > Thank you for your thorough reading of our response, and for taking it into consideration for raising your score, we greatly appreciate it!
> > >
> > > We will definitely implement the changes you suggest, they are very helpful. We will expand on the insights gained by our analysis in Section 2, and specify the data and hypotheses more clearly in the Abstract and Introduction, alongside their conclusions. We will be sure to cite **Lewandowsky, S., & Farrell, S. (2011). Computational modeling in cognition: Principles and practice. Sage Publications**, thank you for pointing us to it.
> > >
> > > **Given the character limitations, we have split up our reply in two parts. What follows is Part 1 of 2 of our reply.**
> > >
> > > - *Thereby, I am still confused why the slot-based models do not work well at all while you are suggesting that more “factorized representations of object state dynamics” will be important. Shouldn’t slot-based models produce such factorized representations?*
> > >
> > > This is a great question, and a point that we want to emphasize more in our revision. The fixed object slots are an example of an object-centric model that is *not* very reusable on novel dynamic scenes. As demonstrated by our dynamically-equipped video foundation models, our work demonstrates that reusability is an important feature of the most successful models on our benchmarks. Therefore, future models could be object-centric so long as they do *not* sacrifice reusability – since as we show in the case of the fixed object-slot models, being object-centric *alone* is not sufficient to guarantee generalization to novel scenes. One idea that might aid in reusability for this model class to better leverage large video datasets like Ego4D, is to have more dynamically updated object slots (such as in material type or number), though more work would need to be done to properly identify what these update rules should be. Our initial results in this domain is precisely why we are working towards more object-based, video foundation models. We will clarify this in Section 6 in relation to these models since this is an important point we do not want to be missed – thank you for bringing it up.
> > >
> > > - *In fact, could you explain to me...what you actually mean by “dense” neurophysiological data?...not quite sure what you mean with “high-throughput human behavioral readouts”.*
> > >
> > > Thank you for asking this clarification, and we apologize for the lack of clarity in using these terms. We will define these terms in the revised Introduction. The terms “dense” and “high-throughput” are synonymous and refer to the fact that the number of comparisons being made in the neural and behavioral data is large (in the thousands). For example, our Mental-Pong neurophysiological dataset is “dense” because it contains almost 2,000 neurons recorded with high temporal precision, so our models not only have to match each of the neurons well, but also across timepoints and 40 held-out conditions (randomized ball positions). In a similar vein, the OCP task (Bear et al. 2021) tests simulations of a wide range of everyday physical phenomena, including rigid and soft-body collisions, stable multi-object configurations, rolling, sliding, and projectile motion, in a realistic 3D environment with 2,000 scenarios per condition, and is therefore more comprehensive than previous behavioral benchmarks so far. Thus, our neural and behavioral datasets are either high-throughput in behavioral comparisons as OCP is (from 100 human subjects across 16,000 scenarios), or in the neural readout (2,000 neurons, held-out timepoints and ball position conditions).
> > >
> > > **Part 2 of 2 of our reply is continued below.**

---

> > > > ### Author Response · Authors · 2023-08-14
> > > > **Reply Part 2 of 2 to Reviewer 6vWo Comment**
> > > >
> > > > **This is Part 2 of 2 of our reply**.
> > > >
> > > > - *I highly recommend discussing the large gap between human-to-human consistency and model-to-human consistency further...Are the models any good at all? Are model differences in terms of model-to-human correlation actually meaningful? I am still not quite sure.*
> > > >
> > > > Definitely, thank you for the suggestion. There are certainly scenarios where models do reasonably well compared to other scenarios.  From our analysis, these especially seem to be rigid body scenarios involving “Support” relations where stacks of objects fall over depending on their shapes and arrangement; “Collide”, where pairs of objects collide depending on their trajectories and placement; and “Roll”, where objects move across a surface either by rolling or sliding. In this scenarios, the best models can attain human consistency scores much higher than what we report in Figure 2C of around 0.6. And across models, as quantified in the new Figure 2D (included in the pdf attached to the Global Response), it seems “Drape”, which involves soft to rigid interactions, is where models universally struggle the most, bringing down the overall consistency reported in Figure 2C. This suggests a concrete future direction as this robustly emerges across our current state-of-the-art models, namely to develop self-supervised loss functions that can leverage temporal relationships to learn more diverse *material properties* might contribute to future improved models. In the revision we will include a new Figure S4, that will mention the quantifications of the error patterns across scenarios for the best models (rather than all models, as is already done in the new Figure 2D). Thank you for the suggestion.
> > > >
> > > > - *With respect to the rather weak model comparison levels reaches, maybe a comparison to similar modeling approaches – say on the level of primary visual areas – would help?*
> > > >
> > > > Yes, this is a great suggestion! To put the current numbers in perspective a bit, our finding of a *single* class of models (dynamically-equipped video foundation models), matching primate DMFC neural responses twice as well as all alternative models is reminiscent of what was initially seen in primate inferotemporal cortex (IT) by Yamins*, Hong*,..., DiCarlo PNAS 2014 (cf. their Figure 1B), where CNNs are the only class of models that jump to 50% explained variance from the 25-30% explained variance of prior models. At the time, their result was considered a breakthrough, since task-optimized CNNs were very different even paradigmatically from the alternative (more hand-crafted) models they tried. In our case, we see reusability across more general Embodied AI tasks, as one step forward that we are currently able to find towards better goal-driven models of higher-cognitive areas, compared to the current models that are trained on static scene datasets or with bespoke latent representations. In other words, matching neural data can be quite constraining, with the best models that double neural predictivity usually being very different from what came before. But still, there is interesting work to be done, especially within our latent future prediction framework, which our work points to, namely to further learn *reusable*, object-centric (and material property) representations. We definitely agree with the reviewer about the timeliness of our work. We will mention this comparison to prior results in visual cortex in the Discussion section of the revision.
> > > >
> > > > - *Thank you for clarifying the oracle value in the pong task – the ball is not visible behind the occluder – all considered models fail to track the ball, correct?*
> > > >
> > > > Thank you for the question. No, there are definitely models that track the ball. In fact, the models that best track the ball, namely the dynamically-based video foundation models, not only approach DMFC’s own ability to track the ball (cf. Figure 3C, rightmost red and blue bars vs. horizontal grey line), but also overall match neural responses best (cf. Figure 3D, top right circles).

---

> > > > ### Comment · Reviewer_6vWo · 2023-08-14
> > > > **thank you - and few further thoughts**
> > > >
> > > > Thank you also from my side for taking the time to write these very informative responses. I highly appreciate it – nice to see that this rebuttal principle actually works sometimes ;-)
> > > >
> > > > May I point out a few newer papers on object-centric representations: In particular, the paper by
> > > > Yuan, J., Chen, T., Li, B., & Xue, X. (2023). Compositional Scene Representation Learning via Reconstruction: A Survey. arXiv eprint 2202.07135 gives a great overview over most current object-centric approaches (indeed crazy seeing this fast progress on this topic).
> > > >
> > > > An even newer paper not included in that review paper is:
> > > >
> > > > Traub, M., Otte, S., Menge, T., Karlbauer, M., Thuemmel, J., & Butz, M. V. (2023). Learning what and where: Disentangling location and identity tracking without supervision. The Eleventh International Conference on Learning Representations. https://openreview.net/forum?id=NeDc-Ak-H_
> > > >
> > > > The only one I know of that implements reusable self-organizing object-centric representations that separate “what” from “where” trained in a fully unsupervised (or self-supervised, that is, autoregressive) manner.
> > > >
> > > >
> > > > About the held-out data: it would be great if you specify further which parts were held-out, that is, if you actually tested for generalization (in fully novel but related data regimes) or rather randomly picked held-out subsets (either one is fine for now – generalization would of course be stronger).
> > > >
> > > > About the tracking ability and its importance: it would be great to make sure it is emphasized that the model that match best can track the ball during occlusion best.
> > > >
> > > > Great to hear about the reminiscence wrt. a previous breakthrough in IT – very interesting to know – and I think it would strengthen the paper even further to highlight this as well.
> > > >
> > > > Looking forward to seeing the final version of this paper – I will make sure to study it in further detail in my lab. I increase the score to full accept. Thank you again for all your efforts and the great work.

---

> > > > > ### Author Response · Authors · 2023-08-14
> > > > > **Reply to Reviewer 6vWo**
> > > > >
> > > > > Thank you so much for your engagement and support! I have really enjoyed this discussion, and thank you for sending these relevant references about object-centric scene learning.
> > > > >
> > > > > - *About the held-out data: it would be great if you specify further which parts were held-out, that is, if you actually tested for generalization (in fully novel but related data regimes) or rather randomly picked held-out subsets (either one is fine for now – generalization would of course be stronger).*
> > > > >
> > > > > This is a great suggestion, and we will clarify this in the revised Introduction. We do both types of generalization. Specifically, in Figure 2 (OCP), models are evaluated on randomly held-out scenarios of Physion. In Figure 3 (Mental Pong Neural Responses), the models are evaluated on the completely novel Pong environment, both in their ability to track the ball and to match neural responses.
> > > > >
> > > > > - *About the tracking ability and its importance: it would be great to make sure it is emphasized that the model that match best can track the ball during occlusion best.*
> > > > >
> > > > > Definitely, we will make sure this is clear in revised Section 5.2, where this result is discussed.
> > > > >
> > > > > - *Great to hear about the reminiscence wrt. a previous breakthrough in IT – very interesting to know – and I think it would strengthen the paper even further to highlight this as well.*
> > > > >
> > > > > Agreed! We will certainly do that in the revised Discussion.
> > > > >
> > > > > - *Looking forward to seeing the final version of this paper – I will make sure to study it in further detail in my lab. I increase the score to full accept. Thank you again for all your efforts and the great work.*
> > > > >
> > > > > Thank you so much for your thorough and thoughtful review!

---

### Official Review · Reviewer_ZwMf · 2023-07-10

**Soundness:** 3 good
**Presentation:** 3 good
**Contribution:** 3 good
**Rating:** 7
**Confidence:** 4

**Summary:**

This manuscript compared several classes of deep-learning based sensory-cognitive models in their ability to predict human behavior and monkey neural responses in tasks that require reasoning about physical relationships based on visual inputs. They find that the models that match best to **neural data** are the ones trained to predict future states of environments in the latent space of pre-trained foundation models that are themselves optimized for dynamic scenes in self-supervised way. Within these models, the ones that are trained on diverse ranges of tasks are the best. On the other hand, the models that matched **human behavior** best were the models with pixel-wise end-to-end prediction for future scenes trained on the same dataset.

Overall, I think this is a timely work to help point out the gap of many self-supervised foundation models from the computation potentially being performed in the brain. It emphasizes the fact that pixel-wise prediction overemphasizes details thus losing the ability of extracting useful representation as animal brains can extract. The relative distance from human and neural data across models is useful for AI researchers to explore better models along the "gradient" in model space.


**Strengths:**

The models being compared are comprehensive and represent the states of the art.
The metrics being used are reasonable (although with limitation, see my comment for weakness). I appreciate that the authors used correlation of prediction on single stimuli (for human judgement).
The environments that the models are evaluated on focus on physical understanding, which is an important gap for the current AI (e.g., chatGPT and many models lack intuitive physics)
The datasets used for pretraining foundation models also come from a diverse range, from ImageNet to Ego4D and kinetics, especially Ego4D is an ecologically relevant dataset of egocentric videos.


**Weaknesses:**

After reading, although I have obtained a good understanding of the relative performance of the models in predicting physical contact and a somewhat good picture of the relative similarity to human behavioral judgement or monkey neural data in another game (mental-pong), I still lack a good understanding of where the model perform worse and how they are different from the brain. Sure we see the accuracy is lower as in Fig 2A and they are not correlated well enough to human judgement (Fig 2B, sorry that I am describing it as a half-empty bottle), but readers like me are interested in, at least for the best model, how they are worse than humans: is there any pattern among the stimuli that the models turn to make a wrong judgment? Maybe this question is hard to answer as we may lack a good representational space to start with in which we can say the models perform poorly in certain regions, but maybe some examples to put in supplementary material, or some subjective summary from the authors' observation can be helpful, or a more fine-grained comparison between different scenarios (are some scenarios easier for networks? do networks perform more similarly to the brain in some scenarios). Fig2C may provides some hint that a model with better accuracy to start with is more correlated with human, but it still does not answer anything about the pattern of response of a particular model. The analogy of the desire for this aspect of analysis is what psychophysics has done with visual illusions: by analyzing what stimuli cause illusions (a mistake made by the brain) and what models reproduce illusions, we could come up with hypotheses of the computational principle of the brain.

I think some of the metrics being used might potentially mask certain difference between model and human/neural data. For example, the fine-grain comparison of predicted hit-probability in the Physion Object Contact Prediction (OCP) task is based on Pearson correlation. But correlation may not capture the overall variance of prediction. The predicted probabilities across stimuli by a model may exhibit a much bigger or much smaller variance than those of humans while showing the same level of correlation. Models and humans may also exhibit different levels of average probabilities across stimuli which is also not captured by correlation. I think such information as mean variance of prediction may also help understand in what aspects these models are worse than or different from humans.

I do have some worry that the two tasks tested on human and monkeys reflect different aspects of reasoning. The mental-pong task as displayed does not require any 3D reasoning, while the Physion task, depending on the particular stimuli, should require 3D reasoning to some degrees. I worry that the different patterns of performance between Figure 2 and 3 may partly result from this dataset difference.

On the other hand, there is another confound, which is the end-to-end models are used differently in Figure 2 and 3. They were trained and tested on the same environments (but novel scenarios) for Figure 2 but tested on a different environment for Figure 3 (the authors did comment about this second confound in Lines 302-307).

Given the two confounds here, I think the results in Figure 3 are more conclusive than Figure 2 since none of the models being compared were trained on the mental-pong task (the authors did emphasize the result for neural data more than that for the human data in the abstract, which I think is a fair thing to do). But if the authors want to draw some conclusion about model generality (a model trained on a diverse range performs better than those end-to-end models when tested on new tasks), then it seems that an additional comparison in which the end-to-end models are trained on a different dataset than Physion but tested on Physion would be better to be added to Figure 2.


**Questions:**

Maybe I am missing it, but I cannot find whether the fine-grain comparison with human or neural data were performed on held-out data or training data, such as in Fig 2B and 3B,C.
Related to this, I infer from Line 304 that SVF and FitVid were fitted to the same "scenario" of Physion as the scenario being tested on, but Line 301 mentioned they were tested on novel scenario. It might help clarify what exactly “scenario” means, especially given that the same word scenario is also overloaded with the meaning of the 8 different scenarios in the Physion environment.

I understand that there is an urge and publication pressure to make claims such as xx model matches human/neural data. But to me, Fig 2B and 3BC all suggest a non-neglectable gap between the networks and the brain. It is this gap that is more inspiring for future research to fill. I suggest changing some subtitle such as 6. to be a relative statement (e.g., "provide a better match") instead of a binary statement of "can match".



**Limitations:**

Yes. Limitations are well addressed. I want to suggest one additional limitation: a fair comparison between different model architecture/objectives would be training the end-to-end models also on the same data (Ego4D) as those foundation models before the evaluation. Perhaps one reason this cannot be done is due to the limitation of computational resource, which is not the fault of the authors and should not be considered as a negative factor for the decision of this paper, if this is indeed the reason.

---

> ### Author Rebuttal · Authors · 2023-08-08
>
> - *is there any pattern among the stimuli that the models turn to make a wrong judgment? …maybe some examples to put in supplementary material, or some subjective summary from the authors' observation can be helpful, or a more fine-grained comparison between different scenarios (are some scenarios easier for networks?*
>
> One of the main reasons we used correlation-based measures is that 1) they reflect measures used in many prior studies (especially e.g. Bear et al. 2021 where the OCP task is introduced), 2) to be able to compare a large class of models and identify trends across them, and 3) enable comparisons across multiple datasets to understand differences between them (Figure 4A) and promising trends common across them (Figure 4B). Nonetheless, this is a great suggestion, and definitely something that is feasible with the metrics we currently use. In particular, as the reviewer suggests, for the OCP task, we plotted the distribution of per-scenario matches to human error patterns across models. We found a clear pattern emerged, namely that “Drape”, involving a soft material (cloth) draping over other objects by virtue of their shape and the cloth’s material, is generally the hardest scenario for models, whereas “Support”, which involves stacks of objects that may fall over, is the scenario that best matches human error patterns. This is interesting because it suggests that predicting diverse material properties, beyond standard rigid objects (especially those of soft bodies), is a tangible goal for future improved models. We will include this as an additional Figure 2D in the revised manuscript. We also will release our model checkpoints and metrics upon acceptance, so that others may do further analyses on our models. Thank you for making this suggestion.
>
> - *I think some of the metrics being used might potentially mask certain difference between model and human/neural data….But correlation may not capture the overall variance of prediction….I think such information as mean variance of prediction may also help understand in what aspects these models are worse than or different from humans.*
>
> The correlation based measures reflect measures used in prior studies, like Bear et al. 2021 where OCP is introduced, and enable the identification of trends across large classes of models and stimuli on multiple datasets. Moreover, mean variance of prediction would still be computed across the probabilities for each stimuli (e.g. $(1/d) \sum_{i=1}^{d} (x_i - y_i)^2$, where d is the number of stimuli), so we are not sure if it would provide more insight in the direction the reviewer is asking for than correlation. Instead, what might be more in line with what the reviewer is suggesting would be looking across individual classes of stimuli. To this end, we have looked at one such intuitive grouping based on scenario and looking at the distribution of human error pattern matching across stimuli in these scenarios across models, in a new Figure 2D. We found that models struggle most with the “Drape” scenario, where cloth drapes over objects due to shape and material, and align best with human errors in the “Support” scenario involving potentially collapsing stacks of objects. This indicates a concrete target for future models, which should aim to better predict diverse material properties, especially for soft bodies. While beyond the scope of our present study, which is more focused on identifying trends of successful models as a starting point for understanding human and animal intuitive physics, there may be other groupings across stimuli that could reveal additional insight; therefore, we will release our model checkpoints upon acceptance so that others may study further measures.
>
> - *I worry that the different patterns of performance between Figure 2 and 3 may partly result from this dataset difference.*
>
> Both humans and animals can perform Mental-Pong (as shown in Rajalingham et al., 2022a & b), and humans and macaques display 3D reasoning that ensures their survival in the physical world. These general abilities across environments fall under the umbrella of “intuitive physics”. Therefore, while we agree with the reviewer that we expect different environments to potentially recruit different aspects of these abilities (as we quantify in Figure 4A), we want to identify models that can better match **both** fine-grained measures of human and animal intuitive physics well, which we do in Figure 4B in finding that the dynamically-equipped video foundation models can most reasonably do this across model classes. We will emphasize this important point more in Section 6 of the revised manuscript, where Figure 4 is discussed. Thank you for mentioning it.
>
> - *...end-to-end models...were trained and tested on the same environments (but novel scenarios) for Figure 2 but tested on a different environment for Figure 3 (the authors did comment about this second confound in Lines 302-307).*
>
> The latent future prediction dynamics of all the foundation models were trained on Physion just as the end-to-end models were, and those Physion trained dynamics were evaluated against neural and behavioral data, ultimately outperforming the end-to-end Physion models. Despite our interest, training end-to-end models on datasets larger than Physion exceeds our current computational resources, as evidenced by models like FitVid requiring nearly a month on eight A100 GPUs. To somewhat get around this constraint, we trained the best models, the dynamically equipped video foundation models, on Kinetics 700. This made the encoder and dynamics training datasets different from Physion used in the OCP metrics in Figure 2 (6 hatched bars). The top models, VC-1 and R3M, yielded comparable results to their Physion-trained versions in both human accuracy and neural predictivity. This indicates dataset scale isn’t the sole factor; inductive biases also play a key role. We'll discuss this further in Sec. 4 & Limitations in revision.

---

> > ### Comment · Reviewer_ZwMf · 2023-08-12
> >
> > Thanks a lot for the reply!
> > The finding that Drape is most difficult to model is really interesting! I don't know enough of the field but I would be super interested to learn if the order found in figure 1 of the newly submitted pdf resembles any sequence in which children learn about these scenarios (either in terms of showing interest or having control).
> >
> > Anyhow, I really hope to see this paper published!

---

> > > ### Author Response · Authors · 2023-08-14
> > > **Reply to Reviewer ZwMf Comment**
> > >
> > > Thank you for the kind words! Although we don’t make any particular claims about childhood development (rather about adults), your suggestion is a very interesting one. We are not aware of any studies that specifically point to the progression of understanding about rigid vs. deformable objects. However, for example, one study that may be relevant is **Needham, A., & Baillargeon, R. (1998). Effects of prior experience on 4.5-month old infants’ object segregation. Infant behavior and development, 21(1), 1-24**. This paper discusses how infants’ prior interactions with objects (in this case, a box and a cylinder) shape their understanding of whether they are separate or unified in subsequent experiments. This might suggest that the more complex the behavior of an object (e.g., deformability), the more experiences may be needed to understand it – though of course testing this more directly in children with deformable objects would be of great interest! We will definitely mention it in the revised Discussion.

---

### Author Rebuttal · Authors · 2023-08-08

**Global Response:**

We thank the reviewers for their thorough reviews, helpful suggestions, and overall positive enthusiasm about our work.

**For common reference, the core contributions of our work are:**

**1. Dense neurophysiological data strongly constrains hypotheses:** Overall, we find that structural generalization to novel environments and matching dense neurophysiological data to be a strong constraint on models of physical simulation, and many state-of-the-art machine learning models fail to do both of these (cf. Figures 2B, 3B, 3C, and 4A). Yet, monkeys can perform these tasks without substantial training, suggesting that they are already equipped with the necessary neural foundations for mental simulation.

**2. Latent future prediction as a promising paradigm for mental simulation:** More specifically, mental simulation in primates (humans and monkeys) appears to be primarily relevant to *dynamics* that are trained to predict the future state of the environment in a suitable latent space (cf. dynamically-equipped video foundation models in Figures 3B and 4B).

**3. Pre-training dataset scale is not the only factor:** In particular, this latent space is highly constrained, and dataset scale is *not* simply all you need (cf. Figure S1 for within-architecture neural predictivity comparisons across pretraining datasets of high variation).

**4. Not every latent works:** In fact, this latent space does *not* appear to consist of bespoke object slots or prioritize fine-grained details (e.g. at the level of pixels), or even through supervised tasks on static images (Figure 3B, “End-to-End” and “Image Foundation Models” bars).

**5. Reusable latents on dynamic scenes as neural models:** Rather, the latent mainly has to be *reusable* across *dynamic* scenes. Taken together, our results observe a correspondence between the ability to predict neural and behavioral responses for the mental simulation phenomena, and developing useful representations for Embodied AI more generally (Figure 4B). This is in contrast to the prior emphasis on classic computer vision tasks such as classification, segmentation, etc (cf. low neural predictivity of “Image Foundation Models” bars in Figures 3 and 4B) either optimized in a supervised or self-supervised manner, but which have up until now been standard models of the primate visual cortex (cf. Schrimpf et al., 2018).

**Computational Resources:**

Reviewers appreciated our thorough study, which we deemed essential to have many functionally reasonable instantiations of these hypotheses and to have the neural and behavioral data strongly separate these hypotheses. The resources used are listed in Supplementary Section A in the initial submission. For all models except FitVid and SVG 128 × 128 (SVG trained on 128 × 128 pixel images), a single NVIDIA A100 GPU was sufficient to train it. Our study suggests that resource-intensive models like FitVid aren't the best for intuitive physics as they don't generalize well (Figure S2). Dataset scale isn't the sole key; inductive biases, as seen in latent future predictions, are crucial, and merely training on larger imagesets, even 250 times bigger than ImageNet, doesn't ensure matching neural dynamics (Figure 3B). We'll release model weights post-publication for use.

**Based on reviewer comments, here we list the major changes we plan to make to our submission if it is accepted:**

**1. Text changes:** With the extra page granted for the camera-ready version, we will enhance the Introduction (Section 1) and Discussion (Section 7) to highlight our paper’s core message: most state-of-the-art machine learning models do *not* meet our rigorous neural and behavioral benchmarks for future prediction. Specifically, many models, including those trained on dynamic scenes like pixel-wise predictors and object-slot models, fall short, as illustrated in Figure 3B. Instead, our findings point towards models that future predict latent representations honed through self-supervised pretraining from egocentric viewpoints that humans and animals naturally receive, as opposed to with RL objectives (VIP bars, Figures 3B and 4B) or webscale imagesets (“Image Foundation Models”). Furthermore, it is pivotal to have trained dynamics on these latent representations, as the fixed latents alone are not adequate (compare the leftmost bars in Figure 3B to the remaining bars in each group, for instance).

Additionally, Section 2 will emphasize a bit more how prior CogSci results involving graph neural networks (GNNs) that operate on semantic, non-pixel-level features actually motivates our latent future prediction paradigm which factorizes vision & dynamics, while Sections 5.2 and 6 will further stress the importance of model proficiency across multiple metrics.

**2. Additional Figures:** Based on Reviewer ZwMf’s excellent suggestion, we include an additional Figure 2D, to show a more detailed error pattern that consistently emerges across models, revealing that they best match human error patterns overall on “Support” scenarios (along with other rigid body scenarios), and could be most improved on the “Drape” scenario. This suggests a concrete future direction of improving models to predict diverse material properties, in order to better handle soft body interactions.

Also, based on Reviewers 6vWo and bK8w feedback, we are introducing a new Supplementary Figure S3 to specifically highlight latent layers across various models, which were originally included in Figures 2-4 (leftmost bars in each group). The best fixed latents were generally from self-supervised video foundation models, like VC-1 and R3M, which then when dynamically-equipped, excelled in predicting the environment's future state across our metrics.

**Both new Figures 2D and S3 are included in the one page PDF we have uploaded here.**

**Finally, we reply to each reviewer individually. Due to character limitations, we respond to the major comments of each reviewer.**

---

### Decision · Program_Chairs · 2023-09-21

**Decision:**

Accept (spotlight)

**Comment:**

There was unanimous agreement among reviewers that the work is important because mental simulation is relatively understudied but important for embodied models. There was also unanimous agreement that the work is thorough – systematically evaluating SOTA models against neural and behavioral data. The reviewers asked for clarifications and raised concerns about possible experimental confounds. There was a substantial discussion between the authors and the reviewers and the rebuttal appears to have addressed all the issues that were raised. The paper appears to make a significant contribution in the establishment of foundation models and neuroscience benchmarks for mental simulation and the AC recommends the paper be accepted.